# Riemannian Generative Decoder

**Andreas Bjerregaard**                                                    *anje@di.ku.dk*
*University of Copenhagen*

**Søren Hauberg**                                                          *sohau@dtu.dk*
*Technical University of Denmark*

**Anders Krogh**                                                           *akrogh@di.ku.dk*
*University of Copenhagen*

**Reviewed on OpenReview:** *https://openreview.net/forum?id=vuPMXg1FDT*

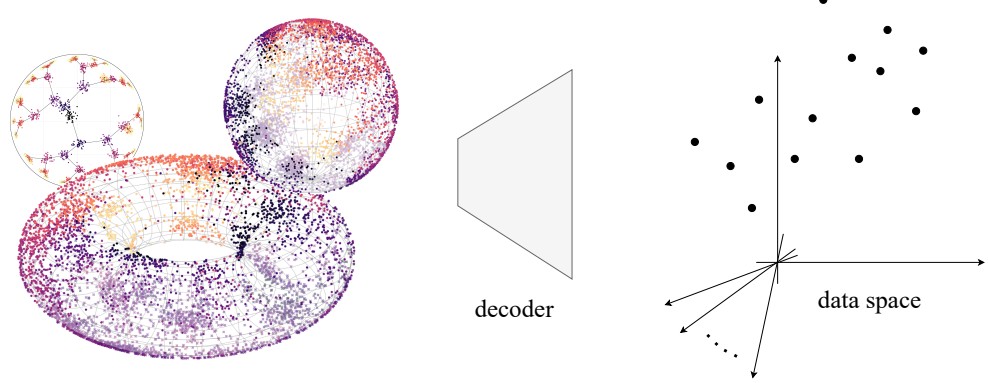

Figure 1: Our decoder reconstructs data from Riemannian manifolds where representations are learned as model parameters via maximum a posteriori.

## Abstract

Euclidean representations distort data with intrinsic non-Euclidean structure. While Riemannian representation learning offers a solution by embedding data onto matching manifolds, it typically relies on an encoder to estimate densities on chosen manifolds. This involves optimizing numerically brittle objectives, potentially harming model training and quality. To completely circumvent this issue, we introduce the *Riemannian generative decoder*, a unifying approach for finding manifold-valued latents on *any* Riemannian manifold. Latents are learned with a Riemannian optimizer while jointly training a decoder network. By discarding the encoder, we vastly simplify the manifold constraint compared to current approaches which often only handle few specific manifolds. We validate our approach on three case studies — a synthetic branching diffusion process, human migrations inferred from mitochondrial DNA, and cells undergoing a cell division cycle — each showing that learned representations respect the prescribed geometry and capture intrinsic non-Euclidean structure. Our method requires only a decoder, is compatible with existing architectures, and yields interpretable latent spaces aligned with data geometry.

🌐 **Home**  yhsure.github.io/riemannian-generative-decoder
🐙 **Code**  github.com/yhsure/riemannian-generative-decoder
🤗 **Data**  hf.co/datasets/yhsure/riemannian-generative-decoder

# 1 Introduction

Real-world data often lie on non-Euclidean manifolds — e.g., evolutionary trees, social-network graphs, or periodic signals — yet most models assume $\mathbb{R}^d$ latent variables. While flexible, this forces data with intrinsic geometrical structure into distorted configurations. Euclidean methods hence often fail to provide visualizations rooted in the geometry which underlies the data, completely missing clear signals (Section 4.3). Meanwhile, low-dimensional projections *directly guide* how practitioners interpret their data in various fields. While non-linear projections like UMAP (McInnes et al., 2018) are widely used despite dangers of misinterpretations (Huang et al., 2022), having more control over the projection facilitates a hypothesis-based exploration of data. For this, *Riemannian manifolds* — spaces that are locally Euclidean but endowed with a smoothly varying inner product (metric) defining lengths, angles, geodesics, and curvature — provide a general framework for modeling geometry. Existing works have adjusted variational autoencoders (VAEs) for embedding data onto various geometries. However, despite the flexibility of VAEs, enforcing manifold priors (e.g., von Mises–Fisher on spheres or Riemannian normals in hyperbolic spaces) requires complex densities and Monte Carlo estimates of normalizing constants, limiting scalability for general manifolds.

We therefore propose the *Riemannian generative decoder*: we discard the encoder and directly learn manifold-valued latents with a Riemannian optimizer while training a decoder network. This encoderless scheme removes the need for approximate densities on the manifold, and handles any Riemannian manifold — including products of heterogeneous manifolds. With a *geometry-aware regularization* through input noise, our model is further encouraged to penalize sharpness relative to the local curvature. We analyze this form of regularization and see its importance in preserving geometric structure during dimensionality reduction. Our contributions are as follows,

- We introduce **a unifying framework for representation learning** on any Riemannian manifold by combining Riemannian optimization with an encoder-less generative model,

- We introduce a **highly scalable *geometric regularization***, promoting coherency between a decoder function and a chosen manifold's metric through noise perturbation,

- We explore various **real-world biological datasets** and find our approach to match or improve a diverse set of metrics; all while being *much stabler* in high dimensions where other methods fail.

# 2 Background

Learned representations often reveal the driving patterns of the data-generating phenomenon. Much of computational biology — and especially data-driven fields like transcriptomics — greatly relies on dimensionality reduction techniques to understand the underlying factors of their experiments (Becht et al., 2019). Unfortunately, a lack of statistical identifiability implies that such representations need not be unique (Locatello et al., 2019). Therefore, it is common practice to inject various inductive biases that reflect prior beliefs or hypotheses about the analyzed problem. One way is to impose a specific geometry on the latent space.

## 2.1 Latent Variable Models

Autoencoders (AEs) learn a deterministic mapping $x \mapsto z \mapsto \hat{x}$ by minimizing a reconstruction loss

$$\min_{\theta,\phi} \sum_{i=1}^{N} L\left(x_i, f_\theta(g_\phi(x_i))\right) \tag{1}$$

where $x_1, \ldots, x_N$ are the training samples, $L$ is the loss function, e.g. the squared error, $g_\phi$ is the encoder and $f_\theta$ the decoder. Because $f_\theta$ is typically smooth, nearby latent codes produce similar reconstructions. This imposes a *smoothness bias* on the representation: distances in latent space are tied to distances in data space.

The variational autoencoder (VAE) by Kingma & Welling (2013) extends this by introducing a prior $p(z)$, a stochastic encoder $q_\phi(z \,|\, x)$ as a variational distribution, and a stochastic decoder $p_\theta(x \,|\, z)$. The marginal

likelihood

$$p(x|\theta) = \int p(x|z,\theta)p(z)dz \tag{2}$$

is intractable, but is lower bounded by the evidence lower bound (ELBO):

$$\log p(x|\theta) \geq \underbrace{\mathbb{E}_{q_\phi(z|x)}\big[\log p_\theta(x \mid z)\big]}_{\text{data reconstruction}} - \underbrace{D_{\mathrm{KL}}\big(q_\phi(z \mid x) \,\|\, p(z)\big)}_{\text{latent regularization}}, \tag{3}$$

where $D_{\mathrm{KL}}$ is the Kullback-Leibler divergence. The decoder is trained by maximizing the ELBO to reconstruct $x$ from samples of $z \sim q_\phi(z \mid x)$. The KL term encourages the encoding distribution to match the prior, typically $\mathcal{N}(0,I)$, while the stochasticity of $q_\phi$ forces the decoder to be robust to perturbations in $z$. Together, these constraints strengthen the smoothness bias across the encoder distribution.

An alternative to the VAE is the Deep Generative Decoder (DGD; Schuster & Krogh 2023), avoiding an encoder entirely. Each latent $z_i$ is treated as a free parameter, and the model uses MAP estimation by maximizing $P(z,\theta,\phi|x)$, corresponding to maximizing the following in $z$, $\theta$ and $\phi$:

$$(\hat{z},\hat{\theta},\hat{\phi}) = \arg\max_{z,\theta,\phi} \sum_{i=1}^{N} \Big( \log p_\theta(x_i \mid z_i) + \log p(z_i \mid \phi) \Big) + \log \big(P(\theta)P(\phi)\big) \tag{4}$$

The last term contains priors on $\theta$ and $\phi$. A parameterized distribution $p(z_i \mid \phi)$ on latent space, such as a Gaussian mixture model, can introduce inductive bias. The decoder smoothness imposes again a continuity constraint on $z \mapsto x$, as reconstructions must interpolate well across learned codes.

Training with the objective in Equation 4 performs MAP estimation of latent codes. Unlike the VAE, it does not maximize a lower bound on the marginal log-likelihood. Regardless of which approximation yields the best inductive bias for generations, our emphasis in the present work is on geometry-aware representation learning. Avoiding encoder-based manifold density approximations is our primary benefit which enables a broad class of latent geometries. We hence treat sampling results mainly as empirical sanity checks.

In all three frameworks — AE, VAE, and DGD — the smoothness of the decoder function acts as a regularizer on latent codes. Since nearby $z$ produce similar outputs, the learned representations *inherit* geometric continuity. The VAE further strengthens this bias through stochastic encodings and KL regularization. The DGD enforces it by directly optimizing per-sample codes under a smooth decoder. These smoothness priors play a central role in learning meaningful low-dimensional structure.

## 2.2 Geometric Inductive Biases

Most learned representations are assumed to be Euclidean. This implies a simple, unbounded topological structure for the representations. This is a flexible and not very informative inductive bias. In scientific settings, however, practitioners often possess explicit knowledge regarding the underlying topology of their data, allowing for direct hypothesis testing via geometric constraints. Selecting a latent manifold is an inductive bias that acts as a constraining regularizer, effectively guiding the learning process towards a more controlled and explainable local minimum.

We briefly survey parts of the literature and generally find that existing approaches involve layers of complexity that potentially limit their performance.

**Spherical representation spaces** encode compactness and periodicity. Davidson et al. (2018) and Xu & Durrett (2018) define latents on $\mathbb{S}^{d-1}$ via a von Mises–Fisher prior:

$$p(z \mid \mu,\kappa) = C_d(\kappa)\exp(\kappa\,\mu^\top z) \quad \text{with} \quad C_d(\kappa) = \frac{\kappa^{d/2-1}}{(2\pi)^{d/2}I_{d/2-1}(\kappa)}, \tag{5}$$

where $\mu \in \mathbb{S}^{d-1}$ and $\kappa > 0$. Sampling uses rejection or implicit reparameterization and KL terms involve Bessel functions, complicating Equation 3 while adding computational overhead and bias.

**Hyperbolic representation spaces** effectively capture hierarchical data structures (Krioukov et al., 2010). A popular choice (used for, e.g., the $\mathcal{P}$-VAE (Mathieu et al., 2019)) is the Poincaré ball $\mathbb{B}^d$, with metric $g_z = \lambda(z)^2 I$, where $\lambda(z) = 2/(1 - \|z\|^2)$, and distance

$$d_{\mathbb{B}}(u, v) = \operatorname{arcosh}\Big(1 + 2\frac{\|u - v\|^2}{(1 - \|u\|^2)(1 - \|v\|^2)}\Big). \tag{6}$$

One typically uses the Riemannian normal prior

$$p(z) \propto \exp\big(-d_{\mathbb{B}}(z, \mu)^2/(2\sigma^2)\big). \tag{7}$$

The ELBO then requires approximating both the intractable normalizing constant of the prior and volume corrections, typically via Monte Carlo or series-expansion methods. Alternative hyperbolic embeddings like the Lorentz (Nickel & Kiela, 2018) or stereographic projections (Skopek et al., 2019) improve computational stability and flexibility but face analogous challenges.

**General geometries** can represent different inductive biases (Kalatzis et al., 2020; Connor et al., 2021; Falorsi et al., 2018; Grattarola et al., 2019). Current literature is based on encoders whose densities generally lack closed-form formulas on arbitrary manifolds $\mathcal{M}$. They rely on approximations like Monte Carlo importance sampling, truncated wrapped normals

$$q(z|\mu, \Sigma) \approx \sum_{k \in \mathbb{Z}^d} \frac{\exp(-\frac{1}{2}\|\operatorname{Log}_\mu(z) + 2\pi k\|_{\Sigma^{-1}}^2)}{(2\pi)^{d/2}|\Sigma|^{1/2}}, \tag{8}$$

or random-walk reparameterization encoders such as $\Delta$VAE (Rey et al., 2019), that simulates Brownian motion using the manifold exponential map: $z = \operatorname{Exp}_\mu(\sum_i \xi_i)$, $\xi_i \sim \mathcal{N}(0, \frac{t}{\text{steps}}I)$.

Related approaches focus on specific geometries or tasks: scPhere (Ding & Regev, 2021) is an application study of spherical and hyperbolic VAEs to biological data, while Nickel & Kiela (2017) learn Poincaré embeddings from supervised relations. Others, like GAGA, assume pre-learned latent structures (Sun et al., 2024). More classical non-linear methods such as Isomap, diffusion maps, or UMAP (Tenenbaum et al., 2000; Coifman & Lafon, 2006; McInnes et al., 2018) visualize intrinsic geometry but do not explicitly learn Riemannian latent variables nor a generative model.

**Curvature regularization.** Independent of encoder–decoder choices, Lee & Park (2023) propose adding explicit intrinsic and extrinsic curvature penalties of the learned manifold. This is similar in nature to one of our contributions. They derive regularizers that depend on second-order derivatives of the decoder — e.g., for intrinsic curvature:

$$\begin{aligned}
\text{IC}_{\text{approx}}(z) = \Big( &\tfrac{1}{2}(w{\cdot}\nabla)\big(w^\top G_f^{-2}(v{\cdot}\nabla)(G_f v)\big) - \tfrac{1}{2}(v{\cdot}\nabla)\big(w^\top G_f^{-2}(v{\cdot}\nabla)(G_f w)\big) \\
&+ \tfrac{1}{4}w^\top G_f^{-3}(v{\cdot}\nabla G_f)(v{\cdot}\nabla)(G_f w) - \tfrac{1}{4}w^\top G_f^{-2}(v{\cdot}\nabla G_f)G_f^{-1}(v{\cdot}\nabla)(G_f w) \\
&- \tfrac{1}{4}w^\top G_f^{-2}(v{\cdot}\nabla G_f)G_f^{-1}(w{\cdot}\nabla)(G_f v) + \tfrac{1}{4}w^\top G_f^{-1}(v{\cdot}\nabla G_f)G_f^{-2}(w{\cdot}\nabla)(G_f v)\Big)^2.
\end{aligned} \tag{9}$$

where $v, w \sim \mathcal{N}(0, I)$, $G_f = J_f(z)^\top J_f(z)$, and $(a{\cdot}\nabla)$ is a directional derivative. Computationally challenging second-order terms enter via $(a{\cdot}\nabla)G_f$ since $\partial J_f$ are Hessian–vector products of $f$. Their objective encourages globally "flat" embeddings in a Riemannian sense. In contrast, our geometry-aware noise induces a first-order Jacobian penalty which aligns local decoder smoothness with the chosen geometry while avoiding challenging computations (Section 3).

## 3 Methodology

We formulate representation learning as a maximum a posteriori (MAP) estimation problem where the latent space is a Riemannian manifold. Much of the difficulty in probabilistically learning representations over non-trivial geometries has been that their densities are notably difficult to work with. Our approach unifies the geometric inductive bias with the generative process, discarding the need for variational approximations. We ultimately build a simple yet effective representation learning scheme that works across different geometries.

### 3.1 Model Formulation

Consider a dataset $X = \{x_i\}_{i=1}^N$ in the data space $\mathcal{X} \subseteq \mathbb{R}^D$. We assume each observation $x_i$ is generated by a corresponding latent variable $z_i$ lying on a smooth $d$-dimensional Riemannian manifold $(\mathcal{M}, g)$. Instead of amortizing inference via an encoder, we treat the latent codes $Z = \{z_i\}_{i=1}^N$ as free parameters to be optimized directly.

We formulate the learning problem as a maximum a posteriori (MAP) estimation. Given a differentiable decoder $f_\theta : \mathcal{M} \to \mathcal{X}$, we jointly optimize the model parameters $\theta$ and the latent codes $Z$ by minimizing the negative posterior:

$$\mathcal{L}(\theta, Z) = \sum_{i=1}^N \Big( -\log p_\theta(x_i|z_i) - \log p(z_i) \Big) - \log p(\theta). \tag{10}$$

The likelihood $p_\theta(x|z)$ models the observation noise; assuming isotropic Gaussian noise in the ambient space yields $\log p_\theta(x|z) \propto -\|x - f_\theta(z)\|_2^2$. The likelihood choice hence defines the reconstruction objective and must be picked according to the nature of the data. The prior $p(z)$ regularizes the latent distribution and enables generation with our model. For compact manifolds, we employ a uniform prior $p(z) = \mathrm{Vol}(\mathcal{M})^{-1}$ with respect to the Riemannian volume measure, resulting in constant $p(z)$. For non-compact manifolds, one may utilize wrapped distributions or Riemannian normals (explained and illustrated in, e.g., Mathieu et al. (2019)).

Optimization of $Z$ is performed directly on the manifold using Riemannian gradient descent. At a point $z \in \mathcal{M}$, the Riemannian metric can be represented in local coordinates by a symmetric positive definite matrix $G(z)$. If $\nabla_z^{\mathrm{E}} \mathcal{L}$ denotes the usual Euclidean gradient of $\mathcal{L}$ with respect to $z$ in these coordinates, then the Riemannian gradient is

$$\nabla_z^{\mathcal{R}} \mathcal{L} \;=\; G(z)^{-1} \nabla_z^{\mathrm{E}} \mathcal{L} \;\in T_z \mathcal{M}, \tag{11}$$

which is the direction of steepest descent measured in the manifold metric.

The update rule for a latent code $z$ at step $t$ is

$$z^{(t+1)} = R_{z^{(t)}} \left( -\eta \, \nabla_{z^{(t)}}^{\mathcal{R}} \mathcal{L} \right), \tag{12}$$

where $R_z : T_z \mathcal{M} \to \mathcal{M}$ is a retraction that maps tangent vectors back to the manifold (e.g., the exponential map) and $\eta$ is the learning rate. This ensures that all iterates $z^{(t)}$ remain on $\mathcal{M}$.

In practice, we minimize Equation 10 by alternating Euclidean updates of $\theta$ with `Adam` (Kingma, 2014) and Riemannian updates of $\mathcal{Z}$ with `RiemannianAdam` (Bécigneul & Ganea, 2018). This latter adaptive variant replaces $-\eta \nabla_{z^{(t)}}^{\mathcal{R}} \mathcal{L}$ by an adaptive search direction $d_t \in T_{z^{(t)}} \mathcal{M}$ before applying the same retraction update. Relying on *geoopt* (Kochurov et al., 2020) for defining tensors and optimizing on manifolds, the implementation becomes exceedingly simple (see Appendix A). Collectively, our setup removes the need for a parametric encoder and the associated complexity of approximating posterior densities on curved spaces.

### 3.2 Geometric Regularization

For manifolds whose metric tensor varies with position, we introduce a *geometry-aware regularization* to inform the model about the metric. During training, each latent $z$ is perturbed with Gaussian noise whose covariance is the *chosen manifold's* inverse Riemannian metric $G^{-1}(z)$. This adapts the noise to local curvature: on homogeneous manifolds such as the hypersphere (where curvature is constant and metric variation merely reflects coordinate scaling) the procedure recovers nearly isotropic noise, whereas on spaces with non-uniform curvature the noise shape is greatly adjusted by location. We outline a derivation inspired by Bishop (1995) and An (1996) to analyze this noise:

Let $\epsilon \sim \mathcal{N}\big(0, \, \sigma^2 G^{-1}(z)\big)$ and define the squared-error loss $L(z) = \big\| f(z, \theta) - y \big\|^2$ for some target $y$. We inject noise to $z$ via the exponential map, which we approximate by the identity to $O(\|\epsilon\|^2)$:

$$z' \;=\; \mathrm{Exp}_z(\epsilon) \;=\; z + \epsilon + O(\|\epsilon\|^2). \tag{13}$$

Ignoring higher order terms $o(\|\epsilon\|^2)$, a second-order Taylor expansion around $z$ gives

$$L(z') \approx L(z) + \nabla_z L(z)^\top \epsilon + \tfrac{1}{2}\,\epsilon^\top \nabla_z^2 L(z)\,\epsilon, \tag{14}$$

Taking expectation over $\epsilon$ and using $\mathbb{E}[\epsilon] = 0$, $\mathbb{E}[\epsilon\epsilon^\top] = \sigma^2\,G^{-1}(z)$, we obtain

$$\mathbb{E}_\epsilon[L(z')] \;=\; L(z) + \tfrac{\sigma^2}{2}\,\mathrm{Tr}\big(\nabla_z^2 L(z)\,G^{-1}(z)\big). \tag{15}$$

For squared error, we have

$$\nabla_z^2 L(z) \;=\; 2\,J(z)^\top J(z) \;+\; 2\sum_k \big(f_k(z) - y_k\big)\,\nabla_z^2 f_k(z), \tag{16}$$

with $J(z) = \partial_z f(z, \theta)$. Following Bishop (1995), we assume that the residual in the second term is usually negligible on average, so substituting back into the expectation gives

$$\mathbb{E}_\epsilon[\,L(z')\,] \;\approx\; L(z) \;+\; \frac{\sigma^2}{2}\,\mathrm{Tr}\Big(2\,J(z)^\top J(z)\,G^{-1}(z)\Big) \tag{17}$$

$$=\; L(z) \;+\; \sigma^2\,\mathrm{Tr}\big(J(z)^\top G^{-1}(z)\,J(z)\big). \tag{18}$$

where the last equality uses cyclicity of the trace. The additive term is the *induced* regularizer from corrupting representations with Gaussian noise of covariance $\sigma^2 G^{-1}(z)$. It penalizes large output gradients weighted by the manifold's predefined inverse metric, aligning decoder smoothness with local curvature. We analyze its effects further in Appendix E. Our concrete implementation mirrors a single Riemannian gradient descent step, but here scaling and retracting a *noise vector* to the manifold rather than a gradient vector (see Listing S2).

### 3.3 Overview of Available Manifolds

The following are manifolds implemented in *geoopt* (Kochurov et al., 2020), applicable for our representation learning. Additional manifolds can readily be added. To allow heterogeneous latent geometries, *geoopt* provides a *ProductManifold* that forms the Cartesian product $\mathcal{M} = \mathcal{M}_1 \times \cdots \times \mathcal{M}_K$ of base manifolds, equipped with the product Riemannian metric $g = g_1 \oplus \cdots \oplus g_K$. This can be used to partition latent coordinates into blocks, each constrained to its own geometry.

- *Euclidean*
- *Stiefel*
- *CanonicalStiefel*
- *EuclideanStiefel*
- *EuclideanStiefelExact*
- *Sphere*
- *SphereExact*

- *Stereographic*
- *StereographicExact*
- *PoincareBall*
- *PoincareBallExact*
- *SphereProjection*
- *SphereProjectionExact*
- *Scaled*

- *ProductManifold*
- *Lorentz*
- *SymmetricPositiveDefinite*
- *UpperHalf*
- *BoundedDomain*

Manifold parameterization and further details appear on `geoopt.readthedocs.io`. For hyperbolic geometries, we use $c > 0$ to denote the magnitude of the sectional curvature, so the sectional curvature is $-c$. In *geoopt*, this matches `PoincareBall(c)`, whereas the Lorentz implementation uses a parameter $k$ satisfying $\langle x, x \rangle_L = -k$, which corresponds to sectional curvature $-1/k$. Thus, a Lorentz model with curvature $-c$ uses $k = 1/c$.

### 3.4 Datasets

**Cell cycle stages.** Measuring gene expression levels of individual fibroblasts with single-cell RNA sequencing captures a continuous, asynchronous progression through the cell division cycle. Transcriptomic changes

occur through these phases, yielding cyclic patterns in gene expression — topologically, this process forms closed loops ($\mathbb{S}^1$) or higher-dimensional tori (Rappez et al., 2020; Rizvi et al., 2017). As the data is not coupled in nature (we cannot identify and keep track of individual cells), unsupervised learning is suitable for picking up patterns about the underlying distribution of cells.

We apply our *Riemannian generative decoder* to the human fibroblast scRNA-seq dataset (5 367 cells $\times$ 10 789 genes) introduced in DeepCycle (Riba et al., 2022) and archived on Zenodo (Riba, 2021). Data were already preprocessed by scaling each cell to equal library size, log-transforming gene counts, and smoothing and filtering using a standard single-cell pipeline. Before modeling, we subsampled to 189 genes annotated with the cell cycle gene ontology term (GO:0007049) retrieved via QuickGO (Binns et al., 2009) in accordance with other cell cycle studies.

**Branching diffusion process.** The synthetic dataset from Mathieu et al. (2019)[1] simulates tree-structured data via a hierarchical branching diffusion: from a root at the origin in $\mathbb{R}^d$ we grow a depth-$D$ tree where each node at depth $\ell$ produces $C$ children by

$$x_{\text{child}} = x_{\text{parent}} + \epsilon, \quad \epsilon \sim \mathcal{N}\left(0, \frac{\sigma_b^2}{p^\ell}I\right). \tag{19}$$

For each node we also generate $S$ noisy sibling observations $x_{\text{obs}} = x_{\text{node}} + \epsilon'$ with $\epsilon' \sim \mathcal{N}\left(0, \frac{\sigma_b^2}{fp^\ell}I\right)$. The dataset comprises all the noisy $x_{\text{obs}}$ and is standardized to zero mean and unit variance. We set $d = 50$, $D = 7$, $C = 2$, $\sigma_b = 1$, $p = 1$, $S = 50$, $f = 8$, yielding 6 350 observations.

**Human mitochondrial DNA.** Human mitochondrial DNA (hmtDNA) is a small, maternally inherited genome found in cells' mitochondria. Its relatively compact size and stable inheritance make it a fundamental genetic marker in studies of human evolution and population structure. A relatively rapid mutation rate has led the genomes to distinct genetic variants, named *haplogroups*, which reflect evolutionary branching events. Since such phylogenetic structure is hierarchical and tree-like, hyperbolic geometry is a natural inductive bias: its volume grows exponentially with radius, matching the growth of branching structures and enabling low-distortion representations of evolutionary relationships (Macaulay et al., 2023)

We retrieved 67 305 complete or near-complete sequences (15 400 – 16 700 bp) from GenBank via a query from MITOMAP (MITOMAP, 2023). Sequences were annotated with haplogroup labels using *Haplogrep3* (Schönherr et al., 2023), leveraging phylogenetic trees from PhyloTree Build 17 (Van Oven, 2015). A sequence was kept if the reported quality was higher than 0.9. In addition to haplogroup classification, *Haplogrep3* identified mutations with respect to a root sequence; here, separate datasets were made using either the rCRS (revised Cambridge reference sequence) or RSRS (reconstructed Sapiens reference sequence). Mutations were then encoded in a one-hot scheme, removing mutations with $\leq 0.05$ frequency, resulting in datasets with shapes $61665 \times 6298$ (rCRS) and $57385 \times 5366$ (RSRS). Appendix C displays further characteristics.

## 4 Results and Discussion

In the following, we treat each dataset to evaluate and discuss applications of unsupervised learning on meaningful geometries.

### 4.1 Cell Cycle Stages

Figure 2 shows latent representations learned with different manifolds on the scRNA-seq data containing an underlying cyclical biological process. While we may have an idea of an explainable global optimum — e.g., a neatly arranged circle following the cell cycle stages — optimization of the neural network does not necessarily follow such an idea. Given a model expressive enough, representations lying in a circle could as well be unrolled or have distinct arcs interchanged without any loss in task accuracy. To compare model fidelity and how well manifold distances correspond to the biological geometry, Table 1 lists reconstruction

---

[1]Available under MIT license

fidelities and correlations of phase distances versus manifold geodesic distances. Here we compared to $\mathcal{S}$-VAE (Davidson et al., 2018) and $\Delta$VAE (Rey et al., 2019); see Appendix B for further details. Euclidean $\mathbb{R}^3$ yields best reconstructions (having more degrees of freedom), while $\mathbb{S}^2$ improves correlation with the geometry. Toroidal embeddings show greater run-to-run variability, likely due to the limited expressivity of learning on circles $\mathbb{S}^1$ embedded in 2D.

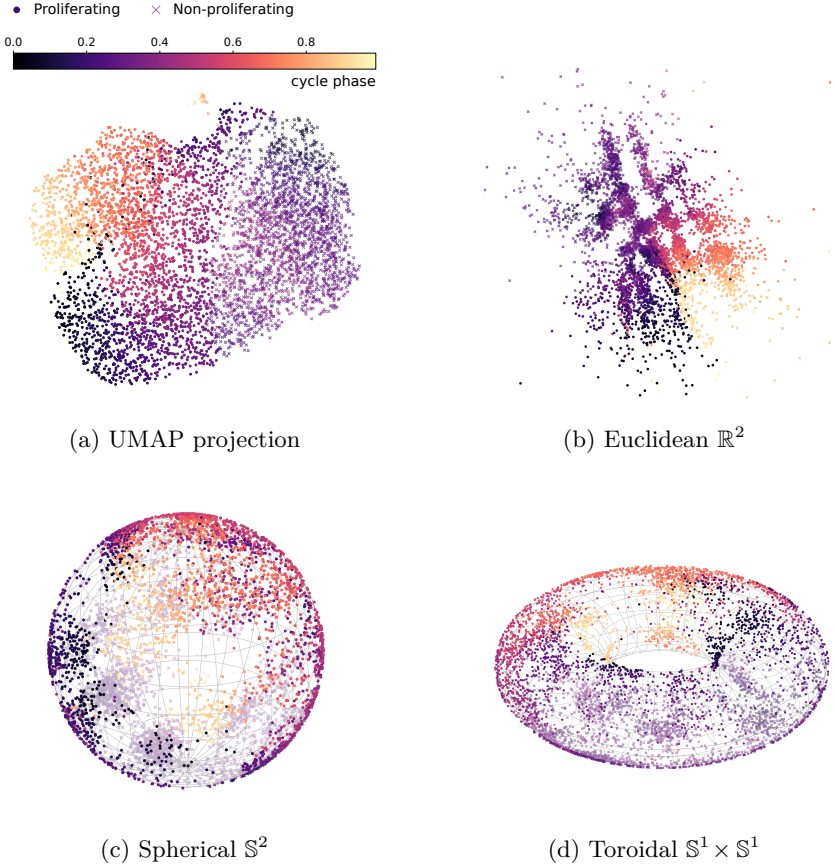

(a) UMAP projection  (b) Euclidean $\mathbb{R}^2$

(c) Spherical $\mathbb{S}^2$  (d) Toroidal $\mathbb{S}^1 \times \mathbb{S}^1$

Figure 2: **Cell cycle phases using either (a) UMAP or (b–d) different Riemannian manifolds.** Samples are concatenated across train/validation/test sets. The phase is inferred by DeepCycle as a continuous variable $\phi \in [0, 1)$ which wraps around such that $\phi = 0$ and $\lim_{\phi \to 1} \phi$ denote the same point in the cycle.

Table 1: **Cell cycle: Correlation and reconstruction metrics across five random initializations** (formatted as mean $\pm$ std). Pearson/Spearman correlate phase distances to latent distances while MAE/MSE measure reconstruction by L1/L2-norm. Our models in gray. Comparisons with $\mathcal{S}$-VAE (Davidson et al., 2018) and $\Delta$VAE (Rey et al., 2019).

| | **Train** | | | | **Test** | | | |
|---|---|---|---|---|---|---|---|---|
| | Pearson | Spearman | MAE | MSE | Pearson | Spearman | MAE | MSE |
| Euclidean $\mathbb{R}^2$ | $0.47_{\pm0.03}$ | $0.50_{\pm0.03}$ | $0.31_{\pm0.00}$ | $0.17_{\pm0.00}$ | $0.52_{\pm0.03}$ | $0.53_{\pm0.03}$ | $0.31_{\pm0.00}$ | $0.18_{\pm0.00}$ |
| Euclidean $\mathbb{R}^3$ | $0.50_{\pm0.05}$ | $0.54_{\pm0.04}$ | $\mathbf{0.30}_{\pm0.00}$ | $\mathbf{0.16}_{\pm0.00}$ | $0.55_{\pm0.04}$ | $0.57_{\pm0.03}$ | $\mathbf{0.31}_{\pm0.00}$ | $\mathbf{0.17}_{\pm0.00}$ |
| Sphere $\mathbb{S}^2$ | $\mathbf{0.58}_{\pm0.03}$ | $\mathbf{0.59}_{\pm0.03}$ | $0.31_{\pm0.00}$ | $0.17_{\pm0.00}$ | $\mathbf{0.60}_{\pm0.03}$ | $\mathbf{0.60}_{\pm0.03}$ | $0.32_{\pm0.00}$ | $0.18_{\pm0.00}$ |
| Torus $\mathbb{S}^1 \times \mathbb{S}^1$ | $0.50_{\pm0.07}$ | $0.51_{\pm0.07}$ | $0.31_{\pm0.00}$ | $0.17_{\pm0.00}$ | $0.52_{\pm0.07}$ | $0.53_{\pm0.08}$ | $0.32_{\pm0.00}$ | $0.18_{\pm0.00}$ |
| $\mathcal{S}$-VAE sphere | $0.50_{\pm0.02}$ | $0.53_{\pm0.03}$ | $0.32_{\pm0.00}$ | $0.19_{\pm0.00}$ | $0.53_{\pm0.02}$ | $0.55_{\pm0.02}$ | $0.32_{\pm0.00}$ | $0.19_{\pm0.00}$ |
| $\Delta$VAE sphere | $0.52_{\pm0.01}$ | $0.55_{\pm0.01}$ | $0.31_{\pm0.00}$ | $0.17_{\pm0.00}$ | $0.57_{\pm0.02}$ | $0.59_{\pm0.02}$ | $0.32_{\pm0.00}$ | $0.18_{\pm0.00}$ |
| $\Delta$VAE torus | $0.43_{\pm0.07}$ | $0.45_{\pm0.07}$ | $0.31_{\pm0.00}$ | $0.17_{\pm0.00}$ | $0.48_{\pm0.06}$ | $0.50_{\pm0.07}$ | $0.32_{\pm0.00}$ | $0.18_{\pm0.00}$ |

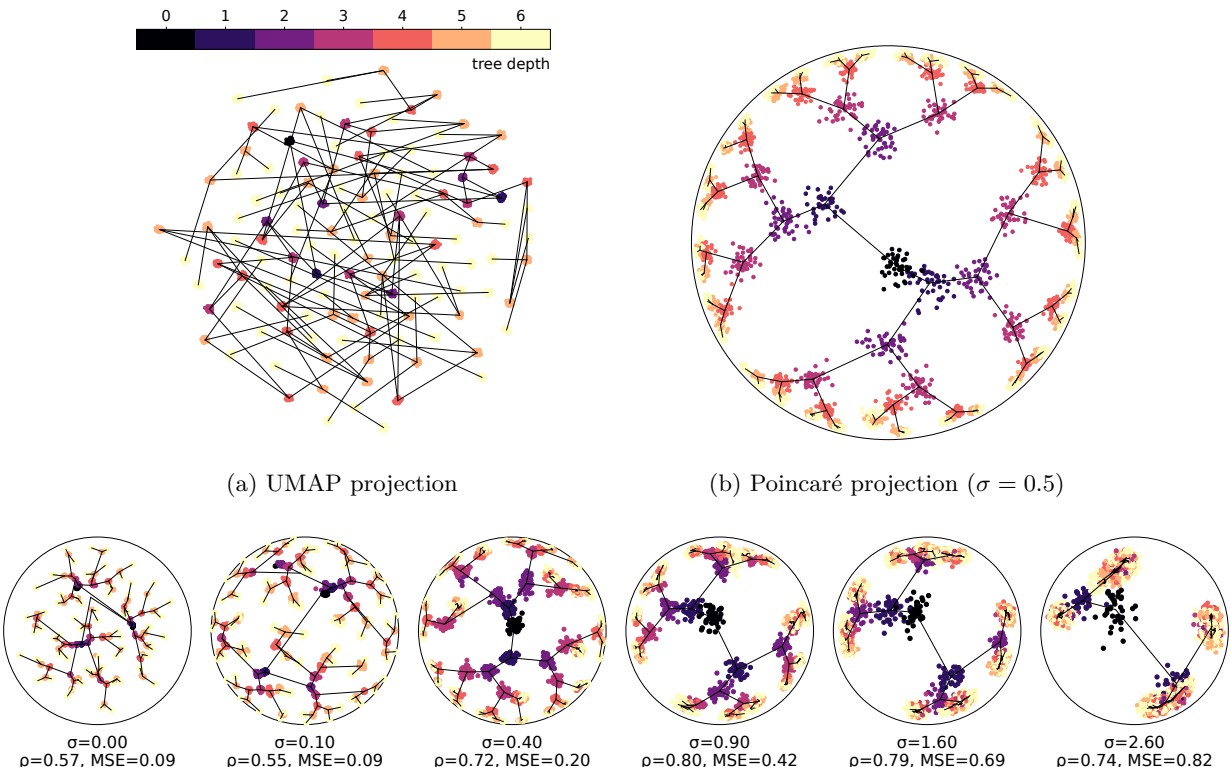

(a) UMAP projection

(b) Poincaré projection ($\sigma = 0.5$)

σ=0.00
ρ=0.57, MSE=0.09

σ=0.10
ρ=0.55, MSE=0.09

σ=0.40
ρ=0.72, MSE=0.20

σ=0.90
ρ=0.80, MSE=0.42

σ=1.60
ρ=0.79, MSE=0.69

σ=2.60
ρ=0.74, MSE=0.82

(c) Progressively increasing regularization noise of the training process (6 distinct models)

Figure 3: **Visualizations of the branching diffusion process.** Trees consist of 7 levels with color lightness denoting depth. Each level adds Gaussian noise (in $d=50$ dimensions) to the previous child with a progressively reduced variance. Visualizations show latents for siblings sampled from the tree nodes. (a) UMAP projection; (b) Poincaré disk projection of Lorentz latents using geometric regularization ($c = 0.2, \sigma = 0.5$); (c) Ablation study showing the influence of the noise scale $\sigma$, listing Pearson correlation $\rho$ and mean squared error on the training set.

## 4.2 Branching Diffusion Process

We find that hyperbolic spaces can efficiently be used as a tool to uncover hierarchical processes. Notably, the UMAP projection (Figure 3a) fails to reveal any underlying tree topology, despite clear cluster separation. In contrast, regularized hyperbolic embeddings in the Poincaré disk (Figure 3b) recover the tree topology. To study the effect of geometric regularization, Figure 3c shows models trained by fixing curvature $c=0.2$ and varying noise from $\sigma = 0$ to $\sigma = 2.6$. Correlations increase sharply up to $\sigma \approx 0.9$, beyond which the noise completely overwhelms the decoder's capacity to preserve pairwise distances. This highlights a tradeoff between preserving local accuracy and enforcing global geometry. Appendix E analyzes and shows how curvature and noise level relate to each other. Figure 4 tracks metric coherency as correlation between manifold versus data-space distance during training; our results show that geometric noise drives correlation steadily higher than using no noise or explicit curvature regularization. This indicates our model succeeds in internalizing the prescribed geometry as an inductive bias.

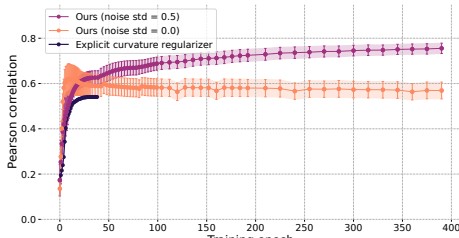

Figure 4: **Pearson correlation of manifold distance versus data-space distance during training** for hyperbolic models on the branching diffusion dataset over 5 runs. The explicit curvature model was stopped after three training days and only shows results from one run (refer to Table 5).

Table 2: **Branching diffusion: Correlation and reconstruction metrics across five initializations** (formatted as mean ± std). Pearson and Spearman correlate all pairs of distances in the tree structure with latent geodesic distances. Our models in gray. Comparison with $\mathcal{P}$-VAE (Mathieu et al., 2019).

| | Train | | | | Test | | | |
|---|---|---|---|---|---|---|---|---|
| | Pearson | Spearman | MAE | MSE | Pearson | Spearman | MAE | MSE |
| *Euclidean* $\mathbb{R}^2$ $(\sigma = 0.0)$ | *0.53*±0.01 | *0.49*±0.01 | ***0.14***±0.00 | ***0.03***±0.00 | *0.52*±0.02 | *0.49*±0.02 | *0.18*±0.01 | *0.06*±0.01 |
| *Sphere* $\mathbb{S}^2$ $(\sigma = 0.0)$ | *0.56*±0.02 | *0.53*±0.03 | ***0.14***±0.00 | ***0.03***±0.00 | *0.55*±0.02 | *0.52*±0.02 | ***0.17***±0.00 | ***0.05***±0.00 |
| Lorentz $\mathbb{H}^2$ $(\sigma = 0.1)$ | 0.52±0.02 | 0.48±0.02 | 0.15±0.00 | 0.04±0.00 | 0.48±0.02 | 0.45±0.03 | 0.19±0.02 | 0.08±0.02 |
| Lorentz $\mathbb{H}^2$ $(\sigma = 0.5)$ | 0.78±0.02 | 0.74±0.03 | 0.32±0.01 | 0.18±0.02 | 0.69±0.03 | 0.69±0.03 | 0.28±0.02 | 0.14±0.02 |
| Lorentz $\mathbb{H}^2$ $(\sigma = 1.0)$ | **0.81**±0.02 | **0.77**±0.02 | 0.49±0.01 | 0.39±0.01 | **0.80**±0.02 | **0.76**±0.02 | 0.36±0.01 | 0.21±0.01 |
| Lorentz $\mathbb{H}^2$ $(\sigma = 2.0)$ | 0.77±0.04 | 0.74±0.05 | 0.68±0.01 | 0.74±0.01 | 0.79±0.09 | 0.73±0.11 | 0.52±0.02 | 0.45±0.03 |
| $\mathcal{P}$-VAE $\mathbb{B}^2$ $(c = 1.2)$ | 0.68±0.03 | 0.54±0.07 | 0.42±0.02 | 0.30±0.02 | 0.68±0.04 | 0.54±0.09 | 0.42±0.02 | 0.31±0.02 |

### 4.3 Tracing Human Migrations

By examining differences in hmtDNA sequences, it is possible to infer patterns of migration, lineage, and ancestry. In Figure 5, we show simplified lineages based on Lott et al. (2013) and Van Oven (2015). The figure shows how UMAP fails to uncover the hierarchical nature (panel *a*) while Euclidean embeddings show slight improvement (panel *b*). Hyperbolic models *clearly recover haplogroup hierarchies* (panels *c–d*), regardless of which reference sequence was used to encode the data. This choice otherwise has a big impact on the actual sequence encodings, preprocessing and filtering (see Appendix C), but models on either set converge at strikingly similar representations when using the same seed (panels *c–d*). The geographical locations of haplogroups strongly correspond to the locations of representations when comparing with migration maps from, e.g., Lott et al. (2013). We show qualitative tree-building results on Figure 6.

Appendix D lists correlation and reconstruction metrics for hmtDNA models (with one-hot data-space distance as a proxy for tree distance). For regularized hyperbolic manifolds we found mainly Spearman correlations to improve, denoting a *non-linear correlation*. Intuitively, the manifold succeeds in capturing the hierarchical branching structure of the haplogroup tree, but absolute path lengths are rescaled by the curvature. Figure 5 strongly suggests that hyperbolic distances better relate to the tree structure than Euclidean ones. In Section 4.4, we treat additional quantitative results using hmtDNA metadata.

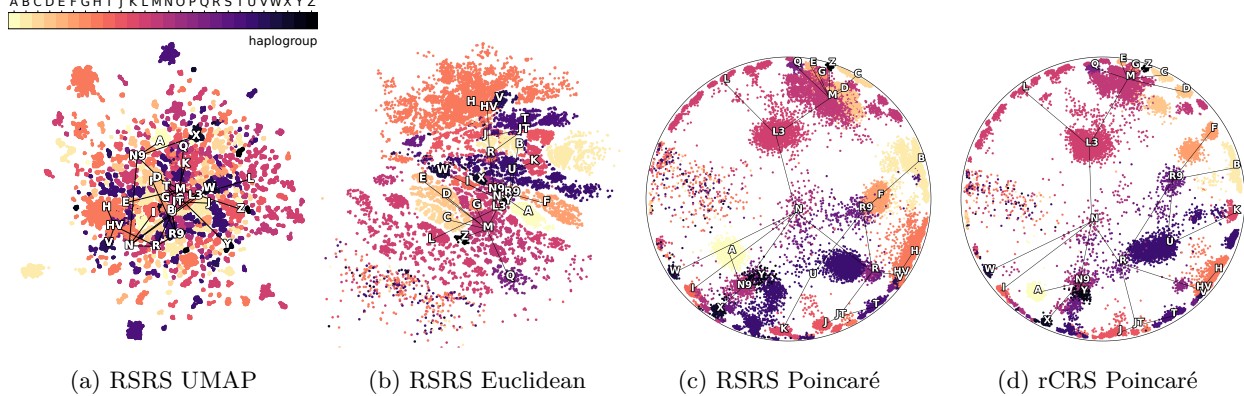

(a) RSRS UMAP   (b) RSRS Euclidean   (c) RSRS Poincaré   (d) rCRS Poincaré

Figure 5: **Visualizations of hmtDNA haplogroups** using either (a) UMAP, (b) Euclidean latent space, or (c–d) Poincaré projection of Lorentz latents ($c = 0.2, \sigma = 0.5$). Edges represent a simplified lineage based on Lott et al. (2013) and Van Oven (2015), nodes indicate median haplogroup positions. Best viewed zoomed in.

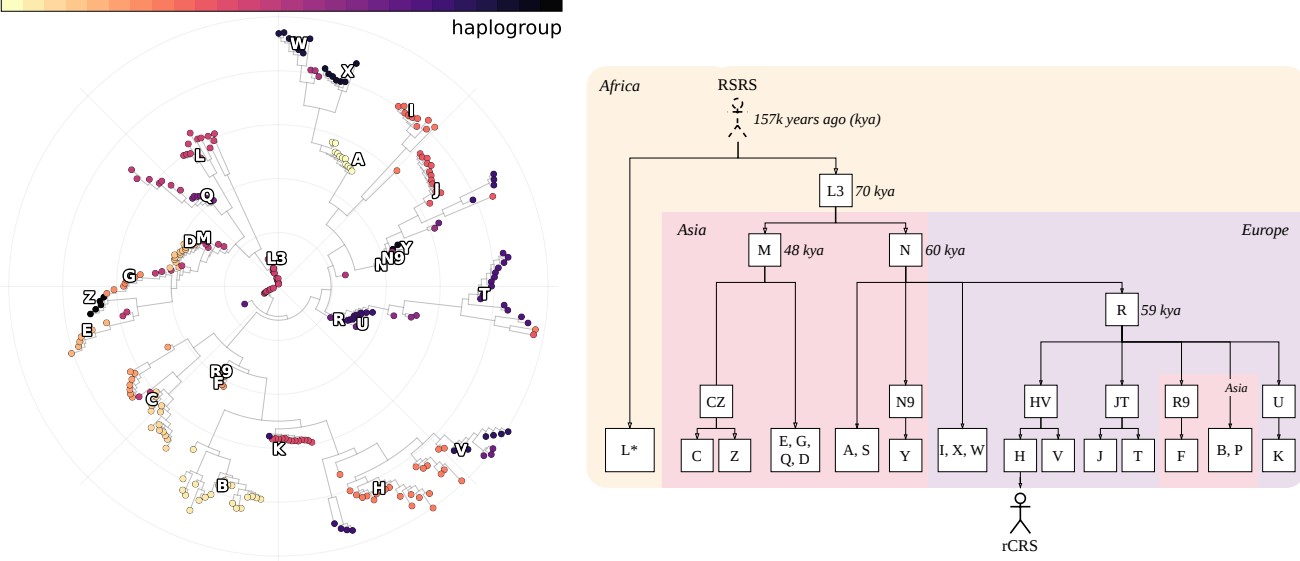

(a) Neighbor joining on pairwise geodesics  (b) Simplified established hmtDNA lineages

Figure 6: **Using geodesics to build trees.** (a) Radial visualization of a *neighbor joining* tree (Saitou & Nei, 1987). This tree is computed from Lorentz distances between 300 sampled hmtDNA representations, producing a data-driven tree with global structure qualitatively similar to the known structure. (b) Simplified diagram of hmtDNA lineages, showing established relationships between haplogroups. The simplified lineages are drawn using plots of Van Oven & Kayser (2009) and Lott et al. (2013).

## 4.4 General Utility

We assess (i) generative fidelity with a discrimination test, (ii) downstream utility from learned latents, and (iii) walltime per epoch. Since many tricks can however increase utility and generative metrics (e.g., training with an adversarial loss or engineering complex decoders), these evaluations act mainly as sanity checks.

**Matches or improves generative performance.** An XG-BoostClassifier (default parameters) is trained to distinguish (1) optimized reconstructions of real test samples versus (2) reconstructions obtained by sampling $z \sim p(z)$ and decoding $p(x \mid z)$. We use half of the cell cycle test set to train the discriminator and the other half to evaluate it; an equal number of synthetic samples is drawn for both splits. Our results indicate that synthetic RGD generations are at least as hard to distinguish from real data reconstructions as generations from VAE baselines (Table 3).

Table 3: **Generative fidelity** measured by XGBClassifier accuracy in discriminating real reconstructions versus synthetic reconstructions (lower is better; perfectly indiscernible gives 0.5 in expectation).

| Manifold | Accuracy |
|---|---|
| RGD Sphere $\mathbb{S}^2$ | **0.58** |
| $\mathcal{S}$-VAE Sphere $\mathbb{S}^2$ | **0.58** |
| $\Delta$VAE Sphere $\mathbb{S}^2$ | 0.62 |
| RGD Torus $\mathbb{S}^1 \times \mathbb{S}^1$ | **0.59** |
| $\Delta$VAE Torus $\mathbb{S}^1 \times \mathbb{S}^1$ | 0.63 |

**Matches or improves downstream utility.** We evaluate downstream performance by fitting both a simple model (logistic/linear regression) and a complex model (XGBClassifier/-Regressor) for classification and regression tasks on latents, respectively. On cell cycle latents, categorical cell stage and continuous cyclic phase yield near-identical scores across methods (Table S2). On hmtDNA latents, classification of geographical region (3-way), haplogroup first letter (24-way), and first two letters (128-way) strongly favor our model with a regularized hyperbolic space (Table 4).

**Unlocks scalability to higher latent dimensionality.** We probed feasibility at higher $d$ on the *full* cell cycle dataset (all genes; typical scRNA-seq analyses use $d \approx 50$). Timing results on one Intel® Xeon®

Table 4: **Human mitochondrial DNA: downstream utility (accuracy)** for logistic regression (LR) or XGBoostClassifier (XGB) on rCRS latents. Trends were consistent for RSRS. Our models in gray.

| Manifold | Region (3-way) | | Haplo 1 (24-way) | | Haplo 2 (128-way) | |
|---|---|---|---|---|---|---|
| | LR | XGB | LR | XGB | LR | XGB |
| Hyperbolic $\mathbb{H}^2_{\sigma=0.1}$ | 0.72 | 0.90 | 0.49 | 0.74 | 0.31 | 0.42 |
| Hyperbolic $\mathbb{H}^2_{\sigma=0.5}$ | **0.86** | **0.97** | **0.70** | **0.85** | **0.43** | 0.41 |
| Euclidean $\mathbb{R}^2$ | 0.69 | 0.85 | 0.46 | 0.74 | 0.31 | **0.43** |
| $\mathcal{P}$-VAE $\mathbb{H}^2$ | 0.52 | 0.65 | 0.19 | 0.44 | 0.13 | 0.41 |

Table 5: **Runtime (s) per epoch** on the cell cycle dataset (all genes) for varying latent dimension (including geometric noise); formatted as mean $\pm$ std through 100 epochs after warmup. Expl. RGD refers to explicit intrinsic curvature regularization (Lee & Park, 2023). *1: Breaks when computing the manifold volume, which involves a term factorial in d. 2: Breaks due to numerical instability (originally ran only at low dimensionality). 3: Computationally challenging (originally ran only at low dimensionality).*

| Latent $d$ | Ours | | | $\Delta$VAE (Eucl.) | $\Delta$VAE (Sphere) | $\mathcal{P}$-VAE (Hyp.) | Expl. RGD (Hyp.) |
|---|---|---|---|---|---|---|---|
| | Eucl. | Sphere | Hyp. | | | | |
| **5D** | **0.21**$_{\pm0.01}$ | **0.23**$_{\pm0.02}$ | **0.25**$_{\pm0.00}$ | 0.41$_{\pm0.02}$ | 0.60$_{\pm0.06}$ | 1.70$_{\pm0.11}$ | 7155$_{\pm174}$[3] |
| **50D** | **0.23**$_{\pm0.03}$ | **0.24**$_{\pm0.01}$ | **0.27**$_{\pm0.00}$ | 0.45$_{\pm0.05}$ | 0.89$_{\pm0.02}$ | breaks[2] | infeasible[3] |
| **500D** | **0.26**$_{\pm0.01}$ | **0.33**$_{\pm0.01}$ | **0.39**$_{\pm0.01}$ | 0.70$_{\pm0.03}$ | breaks[1] | breaks[2] | infeasible[3] |

Gold 6430 core with decoder layers [64, 128, 256] shows stable scaling across manifolds whereas variational baselines scale poorly and become numerically brittle (Table 5).

## Conclusions and Future Directions

We introduced a unifying framework for representation learning on any Riemannian manifold by combining Riemannian optimization with an encoder-less generative model. This simplifies learning since we avoid density estimations, challenging for a general setting of manifolds. With a novel geometric regularization based on noise perturbation, our empirical validations demonstrated our model to successfully capture intrinsic geometric structures across diverse datasets, substantially improving correlations between latent distances and ground truth geometry. While we studied simple, low-dimensional manifolds in an exploratory setting, our method unlocks higher latent dimensionality as well as heterogeneous manifold combinations, notoriously difficult with current methods. While we currently treat the manifold as a simple hyperparameter, future research directions include automatic manifold selection, adaptive geometric regularization strategies, extensions to manifold-valued network weights, and exploring latent manifold structures within pretrained neural networks (e.g., generative diffusion processes or progressive generations from language models).

The decoder-only framework stores each representation explicitly, yielding memory that grows linearly with dataset size. This per-sample parameterization may be prohibitive for datasets of millions of points. Hybrid schemes — such as amortized inference or low-rank factorization — could mitigate this. Lastly, our curated hmtDNA dataset invites further empirical studies, including analyses of geographic distances, migration patterns, or distortion-based metrics of the common consensus trees. We have prepared the data to be easily available on huggingface.co/datasets/yhsure/riemannian-generative-decoder.

## Broader Impact Statement

Our method provides a general tool for manifold-based representation learning that can aid exploratory analysis and hypothesis testing in scientific domains. As with other representation learning methods, applications in sensitive settings (e.g., medical or population studies) should be accompanied by appropriate oversight and compliance with relevant data protection and ethical guidelines.

**Acknowledgments**

This work was partly funded by the Novo Nordisk Foundation through the Center for Basic Machine Learning Research in Life Science (NNF20OC0062606) and by the Pioneer Centre for AI (DNRF grant P1). SH was further funded by VILLUM FONDEN through research grant 42062 and by the European Research Council (ERC) under the European Union's Horizon programme (grant agreement 101125993). AK was further funded by the Novo Nordisk Foundation through grants NNF20OC0059939 and NNF20OC0063268.

The authors thank Yan Li, Helen Wong, Valentina Sora, Viktoria Schuster, Adrián Sousa-Poza, Thilde Terkelsen, and additionally multiple poster attendees for exciting and helpful discussions during different phases of the project.

Guozhong An. The effects of adding noise during backpropagation training on a generalization performance. *Neural computation*, 8(3):643–674, 1996.

Etienne Becht, Leland McInnes, John Healy, Charles-Antoine Dutertre, Immanuel WH Kwok, Lai Guan Ng, Florent Ginhoux, and Evan W Newell. Dimensionality reduction for visualizing single-cell data using umap. *Nature biotechnology*, 37(1):38–44, 2019.

Gary Bécigneul and Octavian-Eugen Ganea. Riemannian adaptive optimization methods. *arXiv preprint arXiv:1810.00760*, 2018.

David Binns, Emily Dimmer, Rachael Huntley, Daniel Barrell, Claire O'donovan, and Rolf Apweiler. Quickgo: a web-based tool for gene ontology searching. *Bioinformatics*, 25(22):3045–3046, 2009.

Chris M Bishop. Training with noise is equivalent to tikhonov regularization. *Neural computation*, 7(1): 108–116, 1995.

Ronald R Coifman and Stéphane Lafon. Diffusion maps. *Applied and computational harmonic analysis*, 21 (1):5–30, 2006.

Marissa Connor, Gregory Canal, and Christopher Rozell. Variational autoencoder with learned latent structure. In *International conference on artificial intelligence and statistics*, pp. 2359–2367. PMLR, 2021.

Tim R Davidson, Luca Falorsi, Nicola De Cao, Thomas Kipf, and Jakub M Tomczak. Hyperspherical variational auto-encoders. *arXiv preprint arXiv:1804.00891*, 2018.

Jiarui Ding and Aviv Regev. Deep generative model embedding of single-cell rna-seq profiles on hyperspheres and hyperbolic spaces. *Nature communications*, 12(1):2554, 2021.

Luca Falorsi, Pim De Haan, Tim R Davidson, Nicola De Cao, Maurice Weiler, Patrick Forré, and Taco S Cohen. Explorations in homeomorphic variational auto-encoding. *arXiv preprint arXiv:1807.04689*, 2018.

Daniele Grattarola, Lorenzo Livi, and Cesare Alippi. Adversarial autoencoders with constant-curvature latent manifolds. *Applied Soft Computing*, 81:105511, 2019.

Haiyang Huang, Yingfan Wang, Cynthia Rudin, and Edward P Browne. Towards a comprehensive evaluation of dimension reduction methods for transcriptomic data visualization. *Communications biology*, 5(1):719, 2022.

Dimitris Kalatzis, David Eklund, Georgios Arvanitidis, and Søren Hauberg. Variational autoencoders with riemannian brownian motion priors. *arXiv preprint arXiv:2002.05227*, 2020.

Diederik P Kingma. Adam: A method for stochastic optimization. *arXiv preprint arXiv:1412.6980*, 2014.

Diederik P Kingma and Max Welling. Auto-encoding variational bayes. *arXiv preprint arXiv:1312.6114*, 2013.

Max Kochurov, Rasul Karimov, and Serge Kozlukov. Geoopt: Riemannian optimization in pytorch, 2020.

Dmitri Krioukov, Fragkiskos Papadopoulos, Maksim Kitsak, Amin Vahdat, and Marián Boguñá. Hyperbolic geometry of complex networks. *Physical Review E—Statistical, Nonlinear, and Soft Matter Physics*, 82(3): 036106, 2010.

Yonghyeon Lee and Frank C Park. On explicit curvature regularization in deep generative models. In *Topological, Algebraic and Geometric Learning Workshops 2023*, pp. 505–518. PMLR, 2023.

Francesco Locatello, Stefan Bauer, Mario Lucic, Gunnar Raetsch, Sylvain Gelly, Bernhard Schölkopf, and Olivier Bachem. Challenging common assumptions in the unsupervised learning of disentangled representations. In *international conference on machine learning*, pp. 4114–4124. PMLR, 2019.

Marie T Lott, Jeremy N Leipzig, Olga Derbeneva, H Michael Xie, Dimitra Chalkia, Mahdi Sarmady, Vincent Procaccio, and Douglas C Wallace. mtdna variation and analysis using mitomap and mitomaster. *Current protocols in bioinformatics*, 44(1):1–23, 2013.

Matthew Macaulay, Aaron Darling, and Mathieu Fourment. Fidelity of hyperbolic space for bayesian phylogenetic inference. *PLoS computational biology*, 19(4):e1011084, 2023.

Emile Mathieu, Charline Le Lan, Chris J Maddison, Ryota Tomioka, and Yee Whye Teh. Continuous hierarchical representations with poincaré variational auto-encoders. *Advances in neural information processing systems*, 32, 2019.

Leland McInnes, John Healy, and James Melville. Umap: Uniform manifold approximation and projection for dimension reduction. *arXiv preprint arXiv:1802.03426*, 2018.

MITOMAP. Mitomap: A human mitochondrial genome database, 2023. URL http://www.mitomap.org. Database.

Maximillian Nickel and Douwe Kiela. Poincaré embeddings for learning hierarchical representations. *Advances in neural information processing systems*, 30, 2017.

Maximillian Nickel and Douwe Kiela. Learning continuous hierarchies in the lorentz model of hyperbolic geometry. In *International conference on machine learning*, pp. 3779–3788. PMLR, 2018.

Adam Paszke, Sam Gross, Francisco Massa, Adam Lerer, James Bradbury, Gregory Chanan, Trevor Killeen, Zeming Lin, Natalia Gimelshein, Luca Antiga, et al. Pytorch: An imperative style, high-performance deep learning library. *Advances in neural information processing systems*, 32, 2019.

Luca Rappez, Alexander Rakhlin, Angelos Rigopoulos, Sergey Nikolenko, and Theodore Alexandrov. Deepcycle reconstructs a cyclic cell cycle trajectory from unsegmented cell images using convolutional neural networks. *Molecular systems biology*, 16(10):MSB209474, 2020.

Luis A Pérez Rey, Vlado Menkovski, and Jacobus W Portegies. Diffusion variational autoencoders. *arXiv preprint arXiv:1901.08991*, 2019.

Andrea Riba. Cell cycle gene regulation dynamics revealed by rna velocity and deep learning, 2021. URL https://doi.org/10.5281/zenodo.4719436. Dataset.

Andrea Riba, Attila Oravecz, Matej Durik, Sara Jiménez, Violaine Alunni, Marie Cerciat, Matthieu Jung, Céline Keime, William M Keyes, and Nacho Molina. Cell cycle gene regulation dynamics revealed by rna velocity and deep-learning. *Nature communications*, 13(1):2865, 2022.

Abbas H Rizvi, Pablo G Camara, Elena K Kandror, Thomas J Roberts, Ira Schieren, Tom Maniatis, and Raul Rabadan. Single-cell topological rna-seq analysis reveals insights into cellular differentiation and development. *Nature biotechnology*, 35(6):551–560, 2017.

Naruya Saitou and Masatoshi Nei. The neighbor-joining method: a new method for reconstructing phylogenetic trees. *Molecular biology and evolution*, 4(4):406–425, 1987.

Sebastian Schönherr, Hansi Weissensteiner, Florian Kronenberg, and Lukas Forer. Haplogrep 3-an interactive haplogroup classification and analysis platform. *Nucleic acids research*, 51(W1):W263–W268, 2023.

Viktoria Schuster and Anders Krogh. The deep generative decoder: Map estimation of representations improves modelling of single-cell rna data. *Bioinformatics*, 39(9):btad497, 2023.

Ondrej Skopek, Octavian-Eugen Ganea, and Gary Bécigneul. Mixed-curvature variational autoencoders. *arXiv preprint arXiv:1911.08411*, 2019.

Xingzhi Sun, Danqi Liao, Kincaid MacDonald, Yanlei Zhang, Chen Liu, Guillaume Huguet, Guy Wolf, Ian Adelstein, Tim GJ Rudner, and Smita Krishnaswamy. Geometry-aware generative autoencoders for warped riemannian metric learning and generative modeling on data manifolds. *arXiv preprint arXiv:2410.12779*, 2024.

Joshua B Tenenbaum, Vin de Silva, and John C Langford. A global geometric framework for nonlinear dimensionality reduction. *science*, 290(5500):2319–2323, 2000.

Mannis Van Oven. Phylotree build 17: Growing the human mitochondrial dna tree. *Forensic Science International: Genetics Supplement Series*, 5:e392–e394, 2015.

Mannis Van Oven and Manfred Kayser. Updated comprehensive phylogenetic tree of global human mitochondrial dna variation. *Human mutation*, 30(2):E386–E394, 2009.

Jiacheng Xu and Greg Durrett. Spherical latent spaces for stable variational autoencoders. *arXiv preprint arXiv:1808.10805*, 2018.

# A   Training Details

```
model.z     := init_z(n, manifold) # initialize points on a manifold          1
model_optim := Adam(model.decoder.parameters())                                2
rep_optim   := RiemannianAdam([model.z])                                       3
                                                                               4
for each epoch:                                                                5
    rep_optim.zero_grad()                                                      6
    for each (i, data) in train_loader:                                        7
        model_optim.zero_grad()                                                8
        z    := model.z[i]                                                     9
        z    := add_noise (z, manifold, std) # optional geometric noise        10
        y    := model(z)                                                       11
        loss := loss_fn(y, data)                                               12
        loss.backward()                                                        13
        model_optim.step()                                                     14
    rep_optim.step()                                                           15
```

Listing S1: Pseudocode for training the *Riemannian generative decoder*.

```
def add_noise (manifold, z, std):                                              1
    noise     := sample_normal(shape=z.shape) * std                           2
    rie_noise := manifold.egrad2rgrad(z, noise)                               3
    z_noisy   := manifold.retr(z, rie_noise)                                  4
    return z_noisy                                                             5
```

Listing S2: Pseudocode for adding geometric noise. `egrad2rgrad` takes a Euclidean gradient and maps it to the Riemannian gradient in the tangent space using the inverse metric. `retr` retracts a tangent vector back onto the manifold via the exponential map if a closed form is available, otherwise a first-order approximation. *geoopt* implements both functions for a wide range of manifolds.

# B   Experimental Details

## B.1   Protocols and Reproducibility

**General details.**   Non-overlapping train/validation/test splits were made using 82/9/9 percent of samples for each distinct dataset. Across data modalities, our models all use linear layers with hidden sizes $[16, 32, 64, 128, 256]$, and sigmoid linear units (SiLU) as the non-linearity between layers. Decoder parameters were optimized via *PyTorch* (Paszke et al., 2019) with Adam (learning rate $2 \times 10^{-3}$, $\beta = (0.9, 0.995)$, weight decay $10^{-3}$), a CosineAnnealingWarmRestarts schedule ($T_0 = 40$ epochs) and early-stopped with patience 85, typically resulting in approximately 500 epochs. Representations were optimized with *geoopt*'s RiemannianAdam (learning rate $1 \times 10^{-1}$, $\beta = (0.5, 0.7)$, stabilization period 5). Spherical/toroidal representations used learning rate $4 \times 10^{-1}$ and decoder $\beta = (0.7, 0.9)$. Representations are only updated once an epoch, necessitating larger learning rates and less rigidity via the beta parameters. For Lorentz hyperbolic manifolds, the curvature was fixed at $c = 0.2$ unless stated otherwise; in *geoopt* this corresponds to the Lorentz parameter $k = 5.0$. The cell cycle and branching diffusion models used mean squared error as reconstruction objective while the hmtDNA models used binary cross entropy.

**Initializing latents.** Initialization strategies for traditional models have been tuned for long. We follow a simple yet robust strategy:

- *Before training:* initial guesses consist of a small degree of random noise projected to the origin of the manifold if it exists, otherwise around a random point. Randomly covering the entire manifold is generally not suitable, since latents cannot easily jump. The quality of both latents and decoder parameters naturally affect each other, but training remains stable.

- *After training:* a number of initial guesses for each point are sampled from around the manifold, and we continue optimization from the one with smallest loss. If there are classes or other distinct regions on the latent manifold, sampling each region is a natural approach. This is a fast and simple variant; one may also train shortly in each sampled location before committing to any, or use initial guesses from a lightweight post-trained encoder.

**Test-time RGD latents.** Following the strategy of Schuster & Krogh (2023), test-time representations for our model were found by freezing the model parameters and finding optimal $z$ through maximizing the log-likelihood of Equation 10.

**UMAP parameters.** For both the branching diffusion and hmtDNA UMAPs (Figure 3a and Figure 5a), hyperparameters `n_neighbors=30` and `min_dist=0.99` were used to help promote global structure. For Figure 2a, the UMAP coordinates of the original study were used.

**Comparison details.** We evaluated existing implementations of three baselines: the $\mathcal{P}$-VAE of Mathieu et al. (2019) (based on MIT-licensed `https://github.com/emilemathieu/pvae`), the $\mathcal{S}$-VAE of Davidson et al. (2018) (based on MIT-licensed `https://github.com/nicola-decao/s-vae-pytorch`), and the $\Delta$VAE of Rey et al. (2019) (based on Apache 2.0-licensed `https://github.com/luis-armando-perez-rey/diffusion_vae`). Model architectures were fixed — here, implementations of earlier methods were adjusted to use the same architectural backbone as ours (see the *General details* paragraph) — while hyperparameters were tuned for each model and dataset.

### B.2 Hardware

All experiments were carried out on a Dell PowerEdge R760 server running Linux kernel 4.18.0-553.40.1.el8_10.x86_64. Key components:

- **CPU**: 2 × Intel® Xeon® Gold 6430 (32 cores/64 threads per CPU, 2.1 GHz base)

- **Memory**: 512 GiB DDR5-4800 (8 × 64 GiB RDIMMs)

- **GPUs**: 1 × NVIDIA A30 (24 GB HBM2e; CUDA 12.8; Driver 570.86.15)

Training single-cell and branching diffusion models takes a few minutes on our setup; models on the mitochondrial DNA data train for around 20 minutes.

## C  hmtDNA Data Distributions

Figure S1 shows the distribution of mutation counts for datasets using different reference sequences.

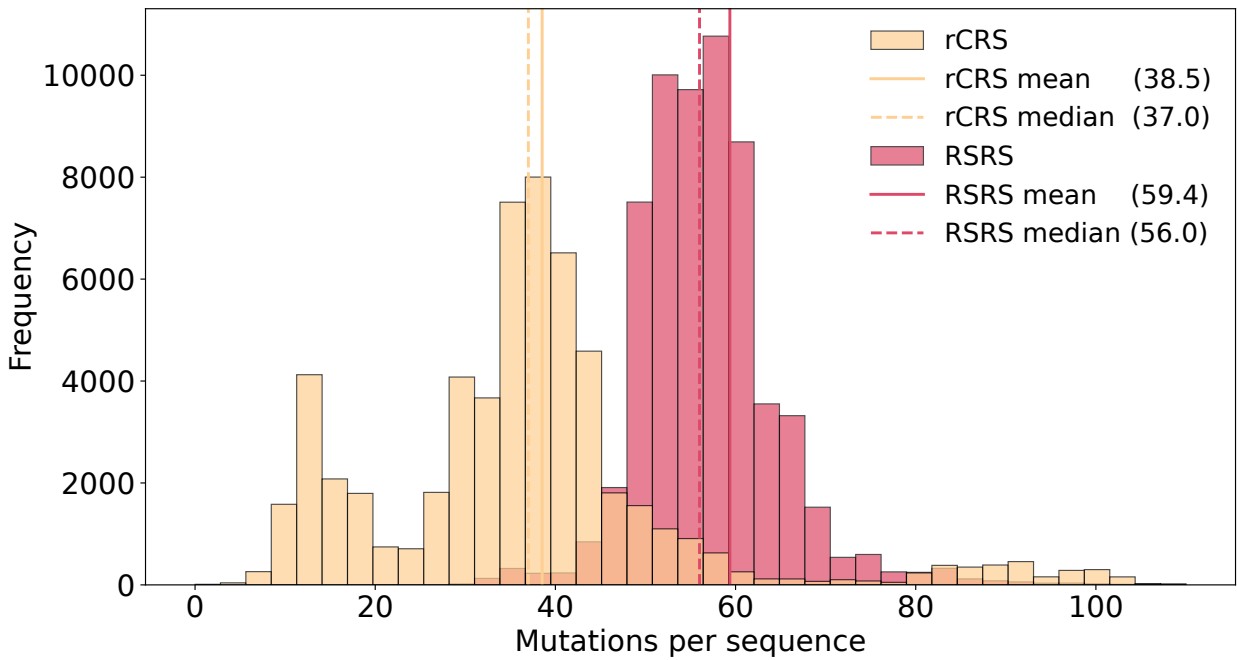

Figure S1: **Distributions of mutation counts** for datasets using different root sequence. Sequence counts of each dataset differ since the choice of haplo-tree changes the reported qualities from *Haplogrep3*, affecting the filtering procedure. Using the revised Cambridge reference sequence (rCRS) means that most sequences contain less mutations when compared to the reconstructed Sapiens reference sequence (RSRS).

## D  hmtDNA Correlations and Reconstructions

In a similar fashion to the other datasets, Table S1 lists correlation and reconstruction metrics for the hmtDNA dataset. It however uses one-hot data-space distance as a heuristic for tree distance.

Table S1: **Correlation and reconstruction metrics across three runs for the hmtDNA dataset** (mean ± std). Mean F1 scores assess reconstruction; Pearson and Spearman correlate manifold vs genetic distance (5000 random points). RSRS/rCRS denote distinct reference sequences.

| | Train | | | Test | | |
|---|---|---|---|---|---|---|
| | Pearson | Spearman | F1 | Pearson | Spearman | F1 |
| $rCRS\ \mathbb{H}^2$ $(\sigma = 0.1)$ | $0.18_{\pm 0.02}$ | $0.17_{\pm 0.05}$ | $0.88_{\pm 0.00}$ | $-0.00_{\pm 0.08}$ | $-0.04_{\pm 0.13}$ | $0.74_{\pm 0.01}$ |
| $rCRS\ \mathbb{H}^2$ $(\sigma = 0.5)$ | $0.28_{\pm 0.01}$ | $\mathbf{0.50}_{\pm 0.04}$ | $0.79_{\pm 0.01}$ | $0.15_{\pm 0.01}$ | $0.28_{\pm 0.03}$ | $\mathbf{0.80}_{\pm 0.01}$ |
| $rCRS\ \mathbb{R}^2$ $(\sigma = 0.0)$ | $\mathbf{0.41}_{\pm 0.03}$ | $0.42_{\pm 0.10}$ | $\mathbf{0.90}_{\pm 0.00}$ | $0.16_{\pm 0.07}$ | $0.24_{\pm 0.14}$ | $0.73_{\pm 0.02}$ |
| $RSRS\ \mathbb{H}^2$ $(\sigma = 0.1)$ | $0.15_{\pm 0.01}$ | $0.12_{\pm 0.04}$ | $0.93_{\pm 0.00}$ | $0.04_{\pm 0.10}$ | $0.04_{\pm 0.23}$ | $0.83_{\pm 0.02}$ |
| $RSRS\ \mathbb{H}^2$ $(\sigma = 0.5)$ | $0.28_{\pm 0.01}$ | $\mathbf{0.49}_{\pm 0.02}$ | $0.86_{\pm 0.00}$ | $0.15_{\pm 0.02}$ | $0.30_{\pm 0.04}$ | $\mathbf{0.88}_{\pm 0.00}$ |
| $RSRS\ \mathbb{R}^2$ $(\sigma = 0.0)$ | $\mathbf{0.35}_{\pm 0.00}$ | $0.29_{\pm 0.01}$ | $\mathbf{0.94}_{\pm 0.00}$ | $0.12_{\pm 0.07}$ | $0.23_{\pm 0.09}$ | $0.83_{\pm 0.02}$ |

# E    Geometric Regularization: Curvature Versus Noise Scale

For a hyperbolic model with curvature $c$, the metric and its inverse carry a nontrivial, state-dependent factor — which cannot be absorbed into a single global $\sigma$.

Concretely, for the ball of curvature $c$ one has

$$G(z) = \frac{4}{(1 - c\,\|z\|^2)^2}\, I, \qquad G^{-1}(z) = \frac{(1 - c\,\|z\|^2)^2}{4}\, I,$$

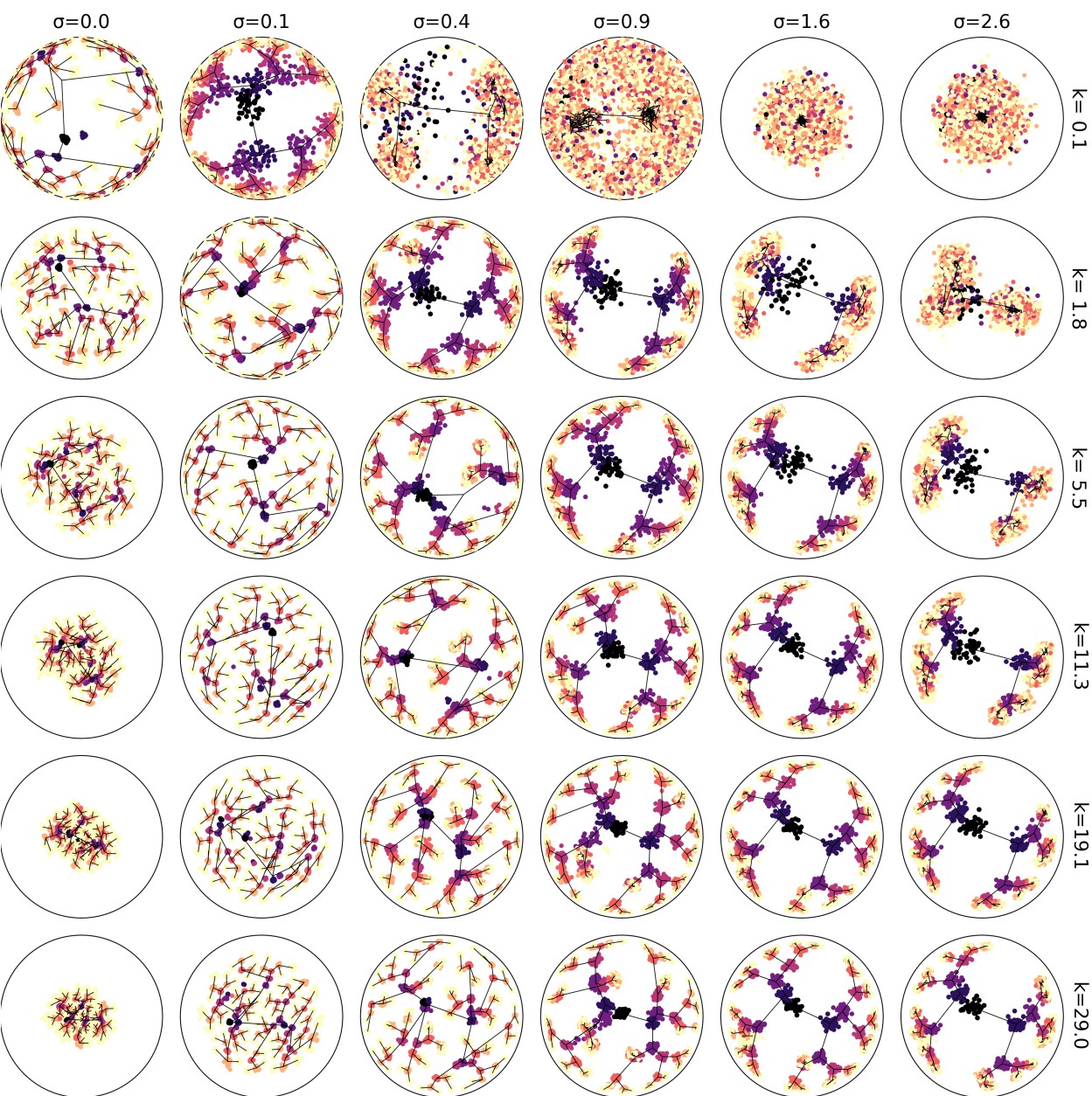

Figure S2: **Effects of manifold curvature and noise level for hyperbolic models on the synthetic branching diffusion dataset**. The visualization is similar to Figure 3c but contains a selection of curvatures rather than $c = 0.2$. Trees consist of 7 levels; color lightness denotes depth.

so the regularizer becomes

$$\mathbb{E}[L(z')] \approx L(z) + \sigma^2 \frac{(1 - c\,\|z\|^2)^2}{4} \operatorname{Tr}\!\big(J(z)^\top J(z)\big).$$

Changing $c$ thus reshapes the weight $\frac{(1-c\,\|z\|^2)^2}{4}$ across the manifold rather than rescaling a uniform noise-variance. Only at $\|z\| \approx 0$ does it reduce to a constant factor, but in general the curvature and noise-scale contribute distinct effects.

The local noise standard deviation induced by this Riemannian scaling is

$$\sigma(z) = \frac{\sigma\,(1 - c\,\|z\|^2)}{2},$$

which depends on both curvature and position. In particular,

$$\frac{\partial \sigma(z)}{\partial c} = -\frac{\sigma\,\|z\|^2}{2} \;<\; 0 \quad (\|z\| > 0),$$

so increasing $c$ *attenuates* the noise magnitude as one moves away from the origin: If one fixes $\sigma$ and increases $c$, then for any $\|z\| > 0$ the factor $(1 - c\,\|z\|^2)$ is smaller, so the actual standard deviation of the injected noise at that point is reduced. Intuitively, points "away from the origin" (larger $\|z\|$) receive less noise. By contrast, raising the global noise scale $\sigma$ amplifies noise uniformly across all $z$. Thus curvature $c$ controls the spatial profile of the perturbations, whereas $\sigma$ governs their overall amplitude. Using the synthetic branching diffusion data, Figure S2 shows the effect of curvature and noise level.

## F    Noise Ablation on the hmtDNA Sequences

Figure S3 shows the effect of geometry-aware regularization, now on the hmtDNA data.

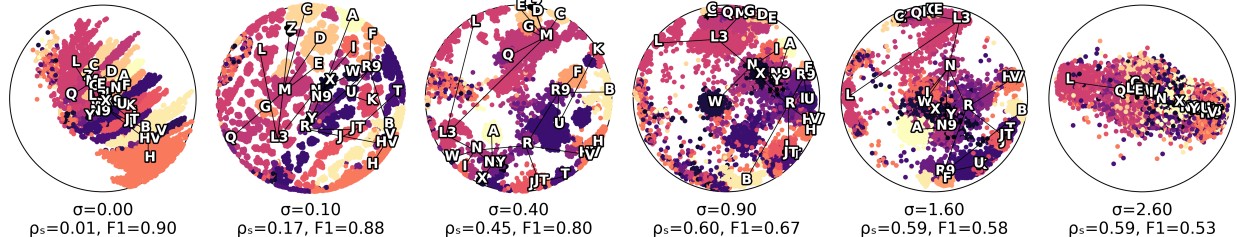

| $\sigma=0.00$ | $\sigma=0.10$ | $\sigma=0.40$ | $\sigma=0.90$ | $\sigma=1.60$ | $\sigma=2.60$ |
| $\rho_s=0.01$, F1=0.90 | $\rho_s=0.17$, F1=0.88 | $\rho_s=0.45$, F1=0.80 | $\rho_s=0.60$, F1=0.67 | $\rho_s=0.59$, F1=0.58 | $\rho_s=0.59$, F1=0.53 |

Figure S3: **Gradually increasing $\sigma$ on the rCRS hmtDNA data**, listing Spearman correlation $\rho_s$ and mean F1-score on the training set. Curvature is $c = 0.2$ corresponding to *geoopt*'s Lorentz-parameter $k = 5.0$.

## G   Relationships of Table 2

Figure S4 visualizes clearly how the noise level $\sigma$ impacts correlation and reconstruction metrics for the synthetic branching diffusion dataset; this figure is based on Table 2.

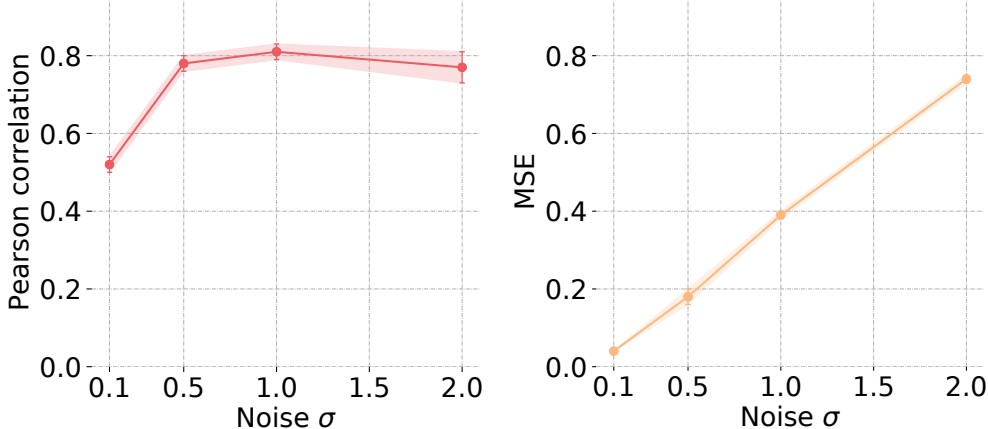

Figure S4: **Errorbar plots varying** $\sigma$ for the geometry-aware regularization; a visualization of the Lorentz results of Table 2.

## H   Cell Cycle Downstream Performance

Table S2 reports downstream performances across methods for predicting categorical cell stage and continuous cell phase from latents.

Table S2: **Cell cycle: downstream utility** — accuracy in predicting cell stage (3-way) or cyclic $R^2$ in regressing cell phase (continuous) using logistic/linear regression (LR) and XGBoost (XGB).

|                                        | Cell stage (3-way) | | Cell phase (cont.) | |
| -------------------------------------- | ---- | ---- | ---- | ---- |
| **Manifold**                           | **LR** | **XGB** | **LR** | **XGB** |
| RGD Euclidean $\mathbb{R}^2$           | 0.89 | 0.90 | 0.41 | 0.86 |
| RGD Euclidean $\mathbb{R}^3$           | 0.93 | 0.91 | 0.44 | 0.87 |
| RGD Sphere $\mathbb{S}^2$              | 0.90 | 0.89 | 0.45 | 0.87 |
| RGD Torus $\mathbb{S}^1 \times \mathbb{S}^1$ | 0.88 | 0.89 | 0.43 | 0.86 |
| $\Delta$VAE Sphere $\mathbb{S}^2$      | 0.90 | 0.90 | 0.47 | 0.88 |
| $\Delta$VAE Torus $\mathbb{S}^1 \times \mathbb{S}^1$ | 0.89 | 0.89 | 0.46 | 0.86 |
| $\mathcal{S}$-VAE $\mathbb{S}^2$       | 0.91 | 0.87 | 0.48 | 0.87 |

