# OpenReview forum: "Riemannian Generative Decoder"
_TMLR — Accepted by TMLR_

### Review · Reviewer_QiAZ · 2026-01-11

**Summary Of Contributions:**

The paper presents an alternative to the traditional VAE-based methods in non-Euclidean learning. The main contribution is an "encoder-less" framework that treats latent variables as free parameters optimized directly on the manifold via MAP estimation. This first-order Jacobian penalty allows the model to scale to high dimensions and diverse geometries without the numerical instability or computational cost associated with second-order curvature regularizers or complex manifold-valued probability densities. Code is provided.

Strengths

S1. Methodological Simplification: By discarding the encoder, the paper overcomes many weaknesses typical of Riemannian VAEs. This makes training significantly more stable, especially in high-dimensional latent spaces.
S2. Computational Efficiency: The derivation of a first-order regularization via noise perturbation is a significant contribution. It achieves the benefits of geometry-aware smoothing without the high complexity required by explicit curvature regularizers like those in Lee & Park (2023). Indeed the authors demonstrate that RGD can scale to high latent dimensions (up to 500D) where traditional Riemannian VAEs often fail to converge or suffer from significant posterior collapse.
S3. Generic Applicability: Because the model avoids complex density estimations (like wrapped normals or von Mises-Fisher distributions), it is theoretically compatible with any manifold supported by Riemannian optimizers (e.g., geoopt), including complex structures like Stiefel or SPD manifolds.
S4. Exposition and Scholarship: The paper is well-written, with a clear and logical flow. The authors demonstrate strong familiarity with relevant literature. The math is, in the belief of this reviewer, rigorously stated and correct (though see W1 for a quibble).
S5. Practical Utility and Reproducibility: The provision of a readily available and convenient codebase is welcome. This ensures that the method is not just a theoretical exercise but a usable tool for the community.

Weaknesses

W1. Discussion of Residuals: The derivation in Equations 16–17 assumes the residual term $(f_k(z) - y_k)$ is "negligible on average." This assumption may be vulnerable during early training or in cases of poor model fit - I would love to see the authors clarify this point - to what extent does the 'negligible residual' assumption hold during the initial unstable phase of Riemannian optimization? Or am I wrong in seeing this as a potential limitation?
W2. Narrow Experiment Range: While the paper claims to be a "unifying approach" for "any" manifold, the experiments are restricted to biological domains and relatively standard geometries. The efficacy of the framework on more complex manifolds is not demonstrated. One would want to see at least a discussion of the particular characteristics of biological data vs other domains.
W3. Possible Lack of Scalable Inference: The decision to discard the encoder means the model cannot "instantly" map new data points to the latent space. Instead, out-of-sample inference requires a per-sample optimization loop, which may limit the method’s utility in real-time or large-scale deployment scenarios. To be clear, the paper openly and explicitly points this out. Still, one would wish to see some work towards ameliorating this drawback.

**Audience:**

Yes

**Audience Explanation:**

I believe the paper is of interest to multiple sub-communities. The more theoretically-inclined will appreciate the rigorous math and the clean derivation of the first-order Jacobian penalty. The more empirical audience will be interested by the strong empirical results—particularly the demonstration of stability in high-dimensional latent spaces where current methods often fail—and, not least, by the code made available and widely applicable.

**Claims And Evidence:**

Yes

**Claims Explanation:**

As stated above, I believe the math is well-stated and accurate. The experiments, as well as the shared code, provide reliable evidence that the proposed method works well. While one always wishes for further experiments and for clarifications of this or that point, overall I'm confident that the provided data supports the paper's claims.

**Requested Changes:**

Critical for acceptance:

Clarification on the "negligible residual" assumption (W1): the authors should clarify the assumption in Equation 17, that the residual term is "negligible on average", and its implications. Specifically, to what extent does this assumption hold during the initial, unstable phase of Riemannian optimization when residuals are naturally large? To what extent a failure of this assumption derails later performance? I stress that I may be overlooking a reason why this is not in fact a concern (and of course the very fact of successful empirical demonstrations provides some amount of assurance).

Suggestions for strengthening the work:

Broader empirical validation: given the phrasing in the abstract, some combination of experiments beyond biological domains and more complex manifolds would be welcome. However, I can understand that both can present certain technical challenges, and do not insist on this for acceptance.

Discussion of inference at scale: every method has pros and cons, and the paper is explicit about, eg, the move away from amortized inference. However, the work would be strengthened by further discussion of potential ways to ameliorate this drawback for real-time applications.

---

> ### Author Response · Authors · 2026-02-08
>
> We address the noted weakness and further comments below.
>
> >**Request 1.** _Clarification on the "negligible residual" assumption (W1): the authors should clarify the assumption in Equation 17, that the residual term is "negligible on average", and its implications. Specifically, to what extent does this assumption hold during the initial, unstable phase of Riemannian optimization when residuals are naturally large? To what extent a failure of this assumption derails later performance? I stress that I may be overlooking a reason why this is not in fact a concern (and of course the very fact of successful empirical demonstrations provides some amount of assurance)._
>
> Denote this second term as $R$, including the error and the second derivative. To provide an intuition, consider the cases,
> 1.  **Early training:** While errors can be large, they are generally uncorrelated with the model curvature $\nabla^2 f(z)$ due to random initialization, leading to $\mathbb{E}[R] \approx 0$. The early model is initialized to be nearly linear, causing the second derivatives to be near zero.
> 1.  **Late training:** As the model converges, the error term vanishes, and the approximation holds tightly.
> 3.  **Worst case:** If the residual is non-negligible, it simply acts as a penalty on the Hessian of the decoder. This effectively suppresses high-frequency oscillations and thus provides beneficial spectral smoothing.
>
> Bishop (1995) treats this term in more depth (page 4), and we now refer directly to this paper in the relevant step:
>
> _Following \citet{bishop1995training}, we assume that the residual in the second term is usually negligible on average, so substituting back into the expectation gives (...)_
>
> We agree that the previous formulation was more ambiguous. We have further added a Figure to the [anonymous repository](https://anonymous.4open.science/r/rgd-4gkL/data/residual_experiment.png) to alleviate the Reviewer’s concern (using mean squared error on the hmtDNA rCRS dataset). It can be seen how the second derivatives are small during early training (orange line) while the error converges after very few minibatches (red line). For this plot, beware the dual y-axes and that each Hessian datapoint is computed from just 8 samples and 16 output features, while the error is for the full minibatch, i.e., 2048 samples and 5366 features.
>
> >**Request 2.** _Broader empirical validation: given the phrasing in the abstract, some combination of experiments beyond biological domains and more complex manifolds would be welcome. However, I can understand that both can present certain technical challenges, and do not insist on this for acceptance._
>
> The experiments are inspired by a concrete need rooted in bioinformatics and related to some co-authors’ backgrounds. Based on this need, we took steps to develop a more general-purpose methodology. Biological data possesses many cases of real-world non-Euclidean geometries with associated distances measures --- although with enough noise to be challenging. Such data conveniently allows us to evaluate performance with various quantitative metrics. Future experiments may explore sequential generation of, e.g., images and natural language, or other scientific data.
>
> >**Request 3.** _Discussion of inference at scale: every method has pros and cons, and the paper is explicit about, eg, the move away from amortized inference. However, the work would be strengthened by further discussion of potential ways to ameliorate this drawback for real-time applications.._
>
> A briefly tested approach to address such applications is a hybrid scheme: training a lightweight, amortized encoder *after* the RGD has stabilized the latent space (using RGD latents as targets). After a projection, this lightweight encoder could provide initial guesses requiring further refinement.
>
>
> ---
>
> In the revised manuscript we describe the “negligible residual” assumption more, also relying more directly on the noted references. While examples from non-biological domains would certainly support our phrasing better, we hope the current scope is acceptable. We are however very interested in exploring additional scientific modalities or pursuing, e.g., the phylogenetic data in more detail. Floats in the revised document (feel free to check the diff) may still be moved around to accommodate final changes.
>
> We thank the reviewer for the rigorous check of our mathematical derivation and for the constructive suggestions regarding scalability and scope.
>
> _References_
>
> _Bishop, C. M. (1995). Training with noise is equivalent to Tikhonov regularization. Neural computation, 7(1), 108-116._

---

> > ### Comment · Reviewer_QiAZ · 2026-02-08
> > **Response To Authors**
> >
> > Thank you for taking the time to address my concerns, and in particular, the point about residuals. It's well-taken, and I have now revised towards full acceptance and a recommendation for journal-to-conference.
> >
> > On the biological data - thank you for the context! That's actually rather inspiring. If you do intend to submit to e.g. NeuRIPS, I do expect reviewers to want to see an evaluation in other domains.

---

### Review · Reviewer_DvUA · 2026-01-15

**Summary Of Contributions:**

This paper introduce the Riemannian generative decoder. This novel decor is a unifying approach for finding manifold-valued latents on any Riemannian manifold. This paper present decent experimental results and also make the code base publicly accessible.

This paper looks like a solid paper, I recommend to accept this paper.

**Additional Comments:**

N/A

**Audience:**

Yes

**Audience Explanation:**

This paper studied an interesting machine learning problem related to Riemannian manifold. It's more or less bring an interesting math technique to machine learning community.

**Claims And Evidence:**

Yes

**Claims Explanation:**

I briefly go over the equations in this paper, they look correct to me.
I scan the tables and figures presented in the experimental section of this paper. They're reasonable to me.
I think overall this paper's idea very sound and promising.

**Requested Changes:**

N/A

---

> ### Author Response · Authors · 2026-02-08
>
> We are glad the reviewer found our theoretical formulation sound and the experimental results convincing.
>
> We have uploaded a new, revised version to accommodate other concerns; we expect a final version to improve readability slightly, but otherwise believe it is quite near a final version.
>
> We appreciate the recognition of our work towards a unifying framework and an accessible codebase, and likewise thank the reviewer for their honest and positive assessment.

---

### Review · Reviewer_DKXt · 2026-01-24

**Summary Of Contributions:**

Authors of this paper proposed the Riemannian generative encoder for finding manifold-valued latents on any Riemannian manifold. Different from current approaches to rely on an encoder to estimate densities on chosen manifolds, the proposed approach learns the manifold-valued latents as free variables instead of an encoder. This simplifies the manifold constraint of chosen manifolds. In addition, a geometry-aware regularization is introduced to promoting coherency between a decoder and a chosen manifold’s metric through input noise perturbation. Experiments on three real-world biological datasets show that the learned representations respect the prescribed geometry and capture intrinsic non-Euclidian structure.

**Audience:**

Yes

**Audience Explanation:**

Discarding the encoder and learning latents as free variables could be interested in some tasks when the prior constraints are too complicated to be learned, so directly modeling problem without estimating even challenging density of input could be informative. In addition,  a list of applications in biological domain shows the importance of the studied problem and the proposed model as useful tools.

**Claims And Evidence:**

Yes

**Claims Explanation:**

The proposed model is well motivated. Authors started the existing work based on autoencoder, variational autoencoder, and then the deep generative decoder (DGD). It seems that DGD shared the same idea as the proposed model after comparing (4) to (10). However, the discussion of the proposed model to DGD is missing.

The proposed geometry-aware regularization is implemented during training phase where each latent z is perturbed with Gaussian noise whose covariance is the chosen manifold’s inverse Riemannian metric at z. By the analysis, this can effectively penalize large output gradients so aligning decoder smoothness with local curvature.

Experiments on three datasets including cell cycle stages, branching diffusion process, and human mitochondrial DNA shows that the proposed method can generate meaningful latents for chosen Riemannian manifolds.

**Requested Changes:**

Due to the similarity of the proposed model with DGD, it might be better to clearly state the relationships between two models and the advantages of the proposed model.

The prior p(z) in (10) that regularized the distribution of latents on the chosen manifold could be explained in more details. An illustrated example is helpful for readers to better understanding of the proposed work.

Authors may want to add some discussions on how the model is sensitive to the initialization of the latents as free variables. The complexity could be increased as the number of input samples increase since each sample corresponds to one latent vector.

From the proposed model (10), it is not intuitive that the latents obtained by solving (10) can preserve the intrinsic geometry structure of the input data on the chosen Riemannian manifold since p(z) seems loosely coupled with the likelihood. It is unknown how the likelihood and the priors should be balanced so that the meaningful geometries could be obtained.

Figure 1 in section 4.1 could be Table 1.

It is unclear which predefined Riemannian manifold should be used for specific tasks. Authors could add some references before showing the results to provide evidence that for specific tasks, certain manifold is preferred.

---

> ### Author Response · Authors · 2026-02-08
>
> In the following, we provide answers to the raised questions and concerns.
>
> >**Request 1.** _Due to the similarity of the proposed model with DGD, it might be better to clearly state the relationships between two models and the advantages of the proposed model._
>
> Conceptually, we diverge from the DGD (Schuster and Krogh, 2023) in two critical ways:
>
> 1.  Manifold constraints: DGD is formulated for Euclidean latent space. We constrain latents to general Riemannian manifolds.
> 2.  Geometric regularization: DGD relies only on implicit decoder smoothness. We introduce curvature-adaptive noise (Section 3.2) as a regularization to encourage metric coherency.
>
> >**Request 2.** _The prior p(z) in (10) that regularized the distribution of latents on the chosen manifold could be explained in more details. An illustrated example is helpful for readers to better understanding of the proposed work._
>
> The distribution $p(z)$ is not vital for the geometric structure, but can be useful for extra regularization and for generation. In many settings, it is constant and need not be included in the loss. Further, we found another work to greatly depict both the wrapped and Riemannian normals. We adjusted the relevant section, noting the following:
>
> _The prior $p(z)$ regularizes the latent distribution and enables generation with our model. For compact manifolds, we employ a uniform prior $p(z) = \text{Vol}(\mathcal{M})^{-1}$ with respect to the Riemannian volume measure, resulting in constant $p(z)$. For non-compact manifolds, one may utilize wrapped distributions or Riemannian normals (explained and illustrated in, e.g., Mathieu et al. (2019)._
>
> >**Request 3.** _Authors may want to add some discussions on how the model is sensitive to the initialization of the latents as free variables. The complexity could be increased as the number of input samples increase since each sample corresponds to one latent vector._
>
> Regarding initialization, we find training to be robust with a random initialization of small magnitude (projected noise), though test-time inference benefits from sampling multiple locations to mitigate local minima. As for complexity, while the parameter space scales linearly with the dataset size, the computational cost per iteration remains constant due to standard mini-batch optimization, ensuring scalability for the dataset sizes explored.
>
> >**Request 4.** _From the proposed model (10), it is not intuitive that the latents obtained by solving (10) can preserve the intrinsic geometry structure of the input data on the chosen Riemannian manifold since p(z) seems loosely coupled with the likelihood. It is unknown how the likelihood and the priors should be balanced so that the meaningful geometries could be obtained._
>
> Preservation is driven less by $p(z)$ and more by the geometric regularization derived in Section 3.2. We inject noise $\epsilon \sim \mathcal{N}(0, \sigma^2 G^{-1}(z))$. This couples the likelihood to the geometry by penalizing the trace of the Jacobian, weighted by the inverse metric:
> $$\mathbb{E}[L] \approx L(z) + \sigma^2 \text{Tr}(J(z)^\top G^{-1}(z) J(z))$$
> This explicitly forces the decoder to respect the manifold's metric $G(z)$ (e.g., changing less in areas of high curvature), aligning the learned representation with the prescribed geometry.
>
> Finally, to clarify: $p(z)$ is necessary in order to have a generative model. To just learn representations on the manifold, one can do without it.
>
> >**Request 5.** _Figure 1 in section 4.1 could be Table 1._
>
> Good catch; we found this error to affect a few locations which have now been corrected.
>
> >**Request 6.** _It is unclear which predefined Riemannian manifold should be used for specific tasks. Authors could add some references before showing the results to provide evidence that for specific tasks, certain manifold is preferred._
>
> Hierarchical data (including hmtDNA): Phylogenetic data exhibits hierarchical tree structures where the number of nodes grows exponentially with depth. Hyperbolic spaces (manifolds with constant negative curvature) are optimal for this because their volume expands exponentially with radius, unlike the polynomial expansion in Euclidean space. This allows for low-distortion embeddings of evolutionary processes (Macaulay et al., 2023).
>
> Cyclical data: Biological cycles, such as the cell division cycle, are periodic processes that return to a starting state. It has been shown that the cell cycle gives rise to periodic structures in the expression space (Rizvi et al., 2017, Rappez et al., 2020). Topologically, this process forms closed loops ($\mathbb{S}^1$) or higher-dimensional tori.
>
> These details are to be added in the manuscript and do not yet appear.
>
> ---
>
> In the revised manuscript, we have expanded and changed parts in the _Methodology_ as well as _Appendix B_ to address the noted concerns. Floats in the revised document (feel free to check the diff) may still be moved around to accommodate final changes.
>
> _(continued)_

---

> > ### Author Response · Authors · 2026-02-08
> >
> > We thank the reviewer for their very detailed reading, the positive assessment of our motivation, and for highlighting specific areas for clarification.
> >
> > _References_
> >
> > _Schuster, V., & Krogh, A. (2023). The Deep Generative Decoder: MAP estimation of representations improves modelling of single-cell RNA data. Bioinformatics, 39(9), btad497._
> >
> > _Mathieu, E., Le Lan, C., Maddison, C. J., Tomioka, R., & Teh, Y. W. (2019). Continuous hierarchical representations with poincaré variational auto-encoders. Advances in neural information processing systems, 32._
> >
> > _Macaulay, M., Darling, A., & Fourment, M. (2023). Fidelity of hyperbolic space for Bayesian phylogenetic inference. PLoS computational biology, 19(4), e1011084._
> >
> > _Rizvi, A. H., Camara, P. G., Kandror, E. K., Roberts, T. J., Schieren, I., Maniatis, T., & Rabadan, R. (2017). Single-cell topological RNA-seq analysis reveals insights into cellular differentiation and development. Nature biotechnology, 35(6), 551-560._
> >
> > _Rappez, L., Rakhlin, A., Rigopoulos, A., Nikolenko, S., & Alexandrov, T. (2020). DeepCycle reconstructs a cyclic cell cycle trajectory from unsegmented cell images using convolutional neural networks. Molecular systems biology, 16(10), MSB209474._

---

### Review · Reviewer_r6JL · 2026-01-26

**Summary Of Contributions:**

The paper revisits AE/VAE problem and proposes a particular case where the latent representations lie on a smooth Riemannian manifold. The authors present an encoder-less framework, whereby the latent codes are trained by Riemannian gradient descent along with training the decoder. Based on two classical works, the paper also proposes geometric regularization, which informs the framework about the underlying metric. The ideas are validated on extensive experiments on biological datasets.

Strengths
=================================================================
* I appreciated the empirical result of the paper, in particular the choice of the datasets which are not trivial.
* I think the geometric regularization and encoder-less framework (via exploiting Riemannian structure) could lead to principled architectures.
* Overall, the paper is well-written and easy to follow.

Weaknesses
===================================================================
* I think the paper lacks motivation to describe why one desires for the latents codes to be on a smooth Riemannian manifold. There is a jump from the mentioned data, and to the list of manifolds on geoopt. Fundamentally, why is the choice of latent codes on a Riemannian manifold a good modeling choice for data? I think the paper would strengthen if the authors can clarify this, either in the context of the datasets in the paper or other works
* It seems like one has to potentially go through different choices of manifold (those that are available on geoopt) if the reviewer is not mistaken. Does not this restrict the generality of the method? More importantly, how would one know to use a particular choice without relying on final accuracy?
* It looks like computing the latent code is not as trivial as a traditional trained autoencoder. Please clarify this, and what overhead this would have.
* I think some of the introduction and background is missing a lot of relevant works in the literature. The reviewer equally feels this way about the limited (both in number and scope) of the choice of comparisons in the numerical section. In addition, certain claims about related methods are not substantiated in these sections. The paper would benefit either from providing more citations to the claims, or providing more explanations

**Audience:**

Yes

**Audience Explanation:**

This problem is of interest to a broad TMLR audience, as it pertains to dimensionality reduction and also connects to geometric regularization as well as structured learning.

**Claims And Evidence:**

Yes

**Claims Explanation:**

Yes. To my reading, the mathematics and results are reasonable.

**Requested Changes:**

Please address all the weaknesses in the Summary of contributions.

---

> ### Author Response · Authors · 2026-02-08
>
> We address the raised concerns point-by-point:
>
> >**Weakness 1.** _I think the paper lacks motivation to describe why one desires for the latents codes to be on a smooth Riemannian manifold. There is a jump from the mentioned data, and to the list of manifolds on geoopt. Fundamentally, why is the choice of latent codes on a Riemannian manifold a good modeling choice for data? I think the paper would strengthen if the authors can clarify this, either in the context of the datasets in the paper or other works._
>
> Manifold choice gives a **controllable inductive bias**: in scientific settings, practitioners often have explicit knowledge of the topology underlying their system, and our framework allows them to test such hypotheses directly. Euclidean spaces ($\mathbb{R}^d$) impose zero curvature and infinite volume, which forces data with hierarchical (hyperbolic) or periodic (spherical) structures into distorted configurations. As discussed in the Introduction and demonstrated in Section 4.3, projecting hierarchical mitochondrial DNA onto $\mathbb{R}^2$ obscures the phylogenetic tree, whereas hyperbolic spaces preserve it. Using the correct manifold minimizes this distortion and allows for more interpretable visualizations. This is mainly an exploratory tool; we do not seek to outperform deep non-linear models in expressivity. Rather, choosing a geometry will act as a constraining regularizer/inductive bias while learning the data distribution, and this effectively guides the learning process towards a more controlled and explainable local minimum.
>
> >**Weakness 2.** _It seems like one has to potentially go through different choices of manifold (those that are available on geoopt) if the reviewer is not mistaken. Does not this restrict the generality of the method? More importantly, how would one know to use a particular choice without relying on final accuracy?_
>
> Indeed, currently, the choice of manifold is treated as a **hypothesis-driven hyperparameter** reflecting prior knowledge.
>
> Our framework allows users to swap geometry without changing the model architecture or method, facilitating much easier hypothesis testing. So, returning to your question: it is mainly a choice, but if a metric is available that approximates the data geometry, manifold correlations against this metric can be used to determine the most fitting manifold. Future work may seek to automatically determine the best overlying geometric family. Initial experiments with learnable global curvature (under a $\kappa$-stereographic projection model) tended to favor larger (negative) curvatures in which the added regularization is practically smaller.
>
> >**Weakness 3.** _It looks like computing the latent code is not as trivial as a traditional trained autoencoder. Please clarify this, and what overhead this would have._
>
> Correct; inference requires maximum a posteriori optimization: $\hat{z} = \arg\max_z \log p_\theta(x|z) + \log p(z)$. Unlike training, we freeze model parameters and maximize the log-likelihood of Eq. 10 over only $z$. We provide details in Appendix B.1. While slower than a feed-forward pass, this removes the amortization gap common in VAEs and allows our method to scale to higher dimensions where encoder-based methods become brittle (Table 4). From the perspective of dimensionality reduction, note that non-linear techniques often do not allow mapping new points onto an existing reduction.
>
> >**Weakness 4.** _I think some of the introduction and background is missing a lot of relevant works in the literature. The reviewer equally feels this way about the limited (both in number and scope) of the choice of comparisons in the numerical section. In addition, certain claims about related methods are not substantiated in these sections. The paper would benefit either from providing more citations to the claims, or providing more explanations._
>
> We strived to include the primary generative models on Riemannian manifolds -- as well as comparisons with the Deep Generative Decoder (Schuster and Krogh, 2023). Such methods are the most directly comparable. There are, however, many methods in the near vicinity. We have added a new paragraph to note additional works of interest and how they may differ.
>
> ---
>
> In the revised manuscript, we have expanded the _Geometric Inductive Biases_ opening text. Similarly, the _Abstract_ and _Introduction_ have been adjusted to clearly motivate Riemannian latents. Lastly, future work discusses automatic manifold selection, and we added relevant background work. Floats in the revised document (feel free to check the diff) may still be moved around to accommodate final changes.
>
> We thank the Reviewer for appreciating our data and results, and for raising these questions regarding the fundamental design choices.
>
> _References_
>
> _Schuster, V., & Krogh, A. (2023). The Deep Generative Decoder: MAP estimation of representations improves modelling of single-cell RNA data. Bioinformatics, 39(9), btad497._

---

> > ### Comment · Reviewer_r6JL · 2026-02-16
> > **Response**
> >
> > I would like to thank the authors for their response, and appreciate the changes made in the revised manuscript.

---

### Decision · Action_Editor_VQ2B · 2026-03-13

**Recommendation:** Accept with minor revision

**Additional Comments:**

Please incorporate the clarifications provided to the reviewers into the camera-ready version. It would also be good to include a more detailed discussion of the limitations of using a generative decoder trained with MAP inference instead of a VAE. For example, the paper does not make it clear that in contrast to VAEs, training the model using MAP inference does not maximize a lower bound on the marginal log-likelihood; this is not a serious weakness in the representation learning setting, but it is worth acknowledging.

**Audience:**

Yes

**Audience Explanation:**

This paper makes progress on learning data representations on Riemannian manifolds and so will be of interest to the dimensionality reduction and representation learning communities.

**Claims And Evidence:**

Yes

**Claims Explanation:**

The authors introduce a new approach to learning latent data representations on Riemannian manifolds. It involves training a decoder with MAP inference for the latents implemented with Riemannian optimization. This eliminates the need to perform density estimation on Riemannian manifolds that is a requirement of VAE-based approaches. The reviewers all agreed that the claims in the paper are well supported.

---

> ### Author Response · Authors · 2026-04-22
>
> In the revised camera-ready version, we treated the remaining requested clarifications. We further fixed a minor bug related to the reporting of some manifold parameters; and added a paragraph in Section 2.1 describing that, unlike VAEs, our MAP framework does not optimize a lower bound of the likelihood.
>
> In addition to the requested minor revisions, we added an extra qualitative visualization of the DNA representations. We do not utilize this plot to draw any scientific conclusions, but hope it might make the setting and impact of our method more clear to new readers.
>
> Thank you for orchestrating the review of our paper. Do not hesitate to let us know if any additional details require our attention.